



# Automated detection of regions with persistently enhanced methane concentrations using Sentinel-5 Precursor satellite data

Steffen Vanselow[1], Oliver Schneising[1], Michael Buchwitz[1], Maximilian Reuter[1],
Heinrich Bovensmann[1], Hartmut Boesch[1], and John P. Burrows[1]

[1]Institute of Environmental Physics (IUP), University of Bremen FB1, Bremen, Germany

**Correspondence:** Steffen Vanselow (vanselow@iup.physik.uni-bremen.de)

**Abstract.** Methane ($CH_4$) is an important anthropogenic greenhouse gas and its rising concentration in the atmosphere contributes significantly to global warming. A comparatively small number of highly emitting persistent methane sources is responsible for a large share of global methane emissions. The identification and quantification of these sources, which often show large uncertainties regarding their emissions or locations, is important to support mitigating climate change. The TRO-

POspheric Monitoring Instrument (TROPOMI) onboard on the Sentinel-5 Precursor (S5P) satellite, launched in October 2017, provides measurements of the column-averaged dry-air mole fraction of atmospheric methane ($XCH_4$) with a daily global coverage and a high spatial resolution of up to $5.5 \times 7 \, \text{km}^2$, enabling the detection and quantification of localized methane sources. We developed a fully automated algorithm to detect regions with persistent methane enhancement and to quantify their emissions using a monthly TROPOMI $XCH_4$ dataset from the years 2018-2021. We detect 217 potential persistent source

regions (PPSRs), which account for approximately $20\,\%$ of the total bottom-up emissions. By comparing the PPSRs in a spatial analysis with anthropogenic and natural emission databases we conclude that $7.8\,\%$ of the detected source regions are dominated by coal, $7.8\,\%$ by oil and gas, $30.4\,\%$ by other anthropogenic sources like landfills or agriculture, $7.3\,\%$ by wetlands and $46.5\,\%$ by unknown sources. Many of the identified PPSRs are well-known source regions, like the Permian Basin in the USA, which is a large production area for oil and gas, the Bowen Basin coal mining area in Australia, or the Pantanal wetlands in

Brazil. We perform a detailed analysis of the PPSRs with the 10 highest emission estimates, including the Sudd Wetland in South Sudan, an oil and gas dominated area on the west coast in Turkmenistan, and one of the largest coal production areas in the world, the Kuznetsk Basin in Russia. The calculated emission estimates of these source regions are in agreement within the uncertainties with results from other studies, but are in most of the cases higher than the emissions reported by emission databases. We demonstrate that our algorithm is able to automatically detect and quantify persistent localized methane sources

of different source type and shape, including larger-scale enhancements such as wetlands or extensive oil and gas production basins.



## 1 Introduction

Methane ($CH_4$) is the second most important anthropogenic greenhouse gas after carbon dioxide ($CO_2$) and its increasing concentration in the atmosphere, which has accelerated in recent years, contributes significantly to global warming (Lan et al.,
2021). Due to its shorter lifetime and higher global warming potential compared to $CO_2$, the reduction of methane emissions can contribute to mitigation of global warming (Shoemaker et al., 2013).

Almost half of the global methane emissions originate from anthropogenic sources, which are dominated by fossil fuel exploitation, livestock, rice cultivation and landfills, whereas the natural emissions mainly originate from wetlands (Saunois et al., 2020). To efficiently reduce methane emissions, a comprehensive understanding of the natural and anthropogenic methane
sources and sinks is required. However, global methane emissions are characterized by large uncertainties, as can be seen in bottom-up inventories which have uncertainties of $20 - 35\%$ for anthropogenic emissions regarding agriculture, fossil fuel and waste and $50\%$ for wetland emissions (Saunois et al., 2020). These uncertainties are strongly related to emissions from individual sources, which are highly uncertain or even partly unknown, especially on regional scale (Saunois et al., 2020). As a result, the explanation of the observed behavior of atmospheric methane, for example during 1998 and 2006 where it
remained at a constant plateau or during the recent years where it showed an accelerated increase, remains challenging. Although, several studies conclude that the recent increase was dominated by an increase in wetland emissions (Lan et al., 2021; Peng et al., 2022; Zhao et al., 2020). In particular, strongly emitting methane sources have a substantial impact on global methane emissions. These include small-scale point sources, so-called super-emitters, such as individual coal mines, natural gas compressor stations or landfills (Lauvaux et al., 2022; Maasakkers et al., 2022; Schuit et al., 2023; Varon et al., 2019).
A comparatively small number of those super-emitters are responsible for a large proportion of methane emissions associated with oil and gas exploitation, coal mining and waste (Frankenberg et al., 2016; Jacob et al., 2016; Lauvaux et al., 2022; Zavala-Araiza et al., 2015). In addition to the super-emitters, larger-scale, but localized source regions also contribute a large share to global methane emissions. These include large oil and gas fields, where smaller sources can emit a huge amount of methane in aggregate, but also regions with high agricultural productivity (rice cultivation, livestock), as well as wetland areas
(Buchwitz et al., 2017; Naus et al., 2023; Pandey et al., 2021; Schneising et al., 2020). The detection and quantification of these small-scale super-emitters and larger-scale source areas is essential in order to assess the contribution of these sources to the global methane emissions and to identify their inherent potential for reducing the global emissions.

Ground-based and aircraft measurements have been used to quantify localized methane sources, but are limited in time and/or space, making (frequent) observations of remote source regions difficult (Borchardt et al., 2021; Frankenberg et al.,
2016; Krautwurst et al., 2021). Satellite measurements, such as from SCIAMACHY (Burrows et al., 1995; Bovensmann et al., 1999) or GOSAT (Kuze et al., 2009, 2016), offer the possibility to globally detect and quantify localized emission sources through temporally frequent global measurements of atmospheric methane (Buchwitz et al., 2017; Jacob et al., 2016, 2022; Sherwin et al., 2024; Thorpe et al., 2023). One important breakthrough in satellite remote sensing of methane in recent years was achieved by the successful launch of the Sentinel-5 Precursor (S5P) satellite in October 2017. Onboard S5P is the TRO-
POspheric Monitoring Instrument (TROPOMI), which is a nadir viewing spectrometer (Veefkind et al., 2012). It provides



observations in the shortwave infrared (SWIR) spectral range with a spatial resolution of $5.5 \times 7 \, \mathrm{km}^2$ from which column-averaged dry air mole fractions of atmospheric methane ($XCH_4$) can be retrieved. Due to the high sensitivity to near-surface concentration changes and the combination of daily global coverage with moderately high spatial resolution, TROPOMI data have already been used to quantify emissions on global and regional scale, including a wide variety of methane sources, such as transient gas leaks, oil and gas fields, coal mining and urban areas, as well as wetland regions (Liu et al., 2021; Naus et al., 2023; Qu et al., 2021; Pandey et al., 2019; Plant et al., 2022; Schneising et al., 2020; Varon et al., 2023; Veefkind et al., 2023). In addition to emission quantification, various studies have shown that TROPOMI can be used to identify point sources on a global scale via plume detection (Lauvaux et al., 2022; Schuit et al., 2023) or via combining with model forecasts (Barré et al., 2021). For example, Barré et al. (2021) created a monitoring methodology to detect $CH_4$ concentration anomalies by comparing TROPOMI data with high-resolution $CH_4$ forecast from the Copernicus Atmosphere Monitoring Service (CAMS). This method can be used to detect missing, underreported and overreported $CH_4$ anomalies in the CAMS data worldwide. Lauvaux et al. (2022) detected methane super-emitters associated with oil and gas production and exploitation for 2019-2020 by analyzing daily TROPOMI data using a plume detection algorithm based on the calculation of local $XCH_4$ enhancements and plume segmentation. The super-emitters were mostly detected over the largest oil and gas basins in Russia, Turkmenistan, USA, Algeria and Middle East and amount to $8-12\%$ of the global oil and gas emissions. Schuit et al. (2023) used TROPOMI data to identify anthropogenic super-emitters including emissions from the sectors coal, oil, gas and landfills for 2021 using a machine-learning approach based on a convolution neuronal network to detect plume-like structures and a support vector classifier to distinguish between real plumes and retrieval artefacts. Methane plumes originating from super-emitters worldwide were identified, mostly from persistent emission clusters, but also from transient sources.

Besides super-emitters, numerous larger-scale strong source regions exist, in which the emissions do not have a plume-like structure as the signals of individual sources within the regions can interfere, which can be the case, for example, in large oil and gas fields (Lauvaux et al., 2022; Naus et al., 2023). Therefore, we developed an automated algorithm to detect and quantify source regions with various sizes, regardless of their source type, including small-scale super-emitters such as coal mine ventilation shafts, but also larger-scale source areas such as wetland areas and large oil and gas fields. We focus on source regions, which show strong and persistent methane enhancements and thus contribute significantly to the global methane emissions. TROPOMI has been providing a vast amount of daily methane data since its launch in 2017. To allow detection of methane source regions in this large dataset on a global scale, we fully automated our detection algorithm. In addition to detection, our algorithm includes a characterization of the source regions, in which the dominant source type is assigned and an emission estimate for each source region is determined.

This study is structured as follows. In section 2, we first present the data that we used for the detection and characterization of the source regions. In Section 3, we describe the algorithm. In Section 4, we present our results, including a global overview of the detected source regions and a detailed analysis of the source regions with the 10 highest emission estimates by comparing our results with emission databases and results from recent studies. At the end, in Section 5, we present our conclusions.





## 2 Data

### 2.1 TROPOMI/WFMD $XCH_4$ data product


The Sentinel-5 Precursor (S5P) with the TROPOspheric Monitoring Instrument (TROPOMI) onboard was launched in October 2017 in a near-polar, sun-synchronous orbit with an equatorial crossing of the ascending node at 13:30 local solar time. TROPOMI is a nadir viewing spectrometer and operates in a push-broom configuration with a swath width of $2600\,\mathrm{km}$, enabling daily global coverage. It measures solar radiation reflected at the earth's surface in the ultraviolet ($267 - 332\,\mathrm{nm}$),

ultraviolet-visible ($305 - 499\,\mathrm{nm}$), near-infrared ($661 - 786\,\mathrm{nm}$) and shortwave infrared ($2300 - 2389\,\mathrm{nm}$) spectral channels (Veefkind et al., 2012). The measurements of TROPOMI in the shortwave infrared (SWIR) spectral range enable the retrieval of column-averaged dry-air mole fractions of atmospheric methane ($XCH_4$) with a horizontal resolution of $5.5 \times 7\,\mathrm{km}^2$ ($7 \times 7\,\mathrm{km}^2$ before August 2019). The radiation back scattered from the earth's surface and measured at the top of the atmosphere have passed through the planetary boundary layer. Therefore, TROPOMI's measurements yield the gas absorption throughout the

atmosphere and importantly close to the earth's surface (Schneising et al., 2019). Consequently, the retrieved $XCH_4$ can be used to detect methane enhancements originating from localized methane sources at the earth's surface.

In this study we use a multi-year (2018-2021) TROPOMI $XCH_4$ dataset retrieved with the Weighting Function Modified Differential Optical Absorption Spectroscopy (WFMD) retrieval algorithm (Buchwitz et al., 2006; Schneising et al., 2011, 2014), which has been adapted and optimized for use on TROPOMI data (Schneising et al., 2019). We use the latest version v1.8 of

the TROPOMI/WFMD product (Schneising et al., 2023) and average the data to monthly $XCH_4$ maps with a spatial resolution of $0.1° \times 0.1°$. In addition to the $XCH_4$, the dataset also includes two variables that are needed for the detection and characterization of the source regions. These variables are: (i) The retrieved surface albedo in the SWIR spectral range and (ii) for each monthly averaged $XCH_4$ grid cell the number of days $N_{\mathrm{days}}$ with TROPOMI measurements from which the monthly mean was calculated. In the following, we refer to this dataset consisting of the $0.1° \times 0.1°$ monthly maps of $XCH_4$, SWIR albedo,

and $N_{\mathrm{days}}$, as $XCH_4$ dataset.

### 2.2 Wind data

Wind data are required to calculate emissions. The European Centre for Medium-Range Weather Forecasts (ECMWF) reanalysis (ERA5) wind product (Hersbach et al., 2020) provides hourly wind data with a horizontal resolution of $0.25° \times 0.25°$ on model levels. From this dataset we computed boundary layer averaged wind speed at the overpass time of TROPOMI. For the

analyzed time period 2018-2021, we computed monthly averaged wind speeds at the same spatial resolution of $0.1° \times 0.1°$ as the $XCH_4$ dataset by calculating the monthly mean wind speed and the standard deviation of the wind speed within the months for each grid cell.



## 2.3 Surface elevation and roughness

The Global Multi-resolution Terrain Elevation Data 2010 (GMTED 2010) is a dataset containing global surface elevation data
available at three different resolutions (approximately 250, 500 and 1,000 m) from various data sources (Danielson and Gesch, 2011). We use the GMTED 2010 to assign the mean surface elevation and the standard deviation of the surface elevation (surface roughness) within the grid cells to the $0.1° \times 0.1°$ grid cells of the $XCH_4$ dataset.

## 2.4 Emission databases

We use the following emission databases to determine the dominant source types of the detected potential source regions by
comparing the emissions of the databases.

### 2.4.1 EDGAR

The Emissions Database for Global Atmospheric Research (EDGAR) v6.0 (Ferrario et al., 2021) is a bottom-up inventory providing detailed information about global anthropogenic emissions of various air pollutants and greenhouse gases. The yearly emission data have a spatial resolution of $0.1° \times 0.1°$ and are available from 1970 to 2018. The emissions base on
data collected from a variety of sources, including international organizations such as the International Energy Agency (IEA), national emission inventories and industry reports. EDGAR provides also sector specific emissions. For methane this includes for example emissions from fossil fuel exploitation, enteric fermentation and rice cultivation.

### 2.4.2 GFEI

The Global Fuel Emission Inventory (GFEI) v2.0 (Scarpelli et al., 2022) is a methane emission database providing global
anthropogenic emissions regarding the fossil fuel sector. The emission data are gridded to yearly maps (2010-2019) with a resolution of $0.1° \times 0.1°$. GFEI uses emission data reported by countries to the United Nations Framework Convention on Climate Change (UNFCCC) and assigns the data to infrastructure locations like coal mines or oil and gas wells. Due to the different approach and thus different emissions compared to EDGAR, GFEI can be used as a useful supplementary database to assign the detected source regions to the corresponding source types.

### 2.4.3 WetCHARTs

WetCHARTs v1.3.1 is a global wetland methane emission ensemble which provides monthly emissions with a resolution of $0.5° \times 0.5°$ for the time period 2001-2019 (Bloom et al., 2021). The ensemble is based on different wetland extent scenarios, multiple terrestrial biosphere models and various temperature dependence parameterizations, resulting in 18 different model configurations. To compare the wetland emissions with the other emission databases, we create a yearly averaged wetland
emission map for 2019 with a resolution of $0.1° \times 0.1°$, by averaging the emissions of all configurations and months.



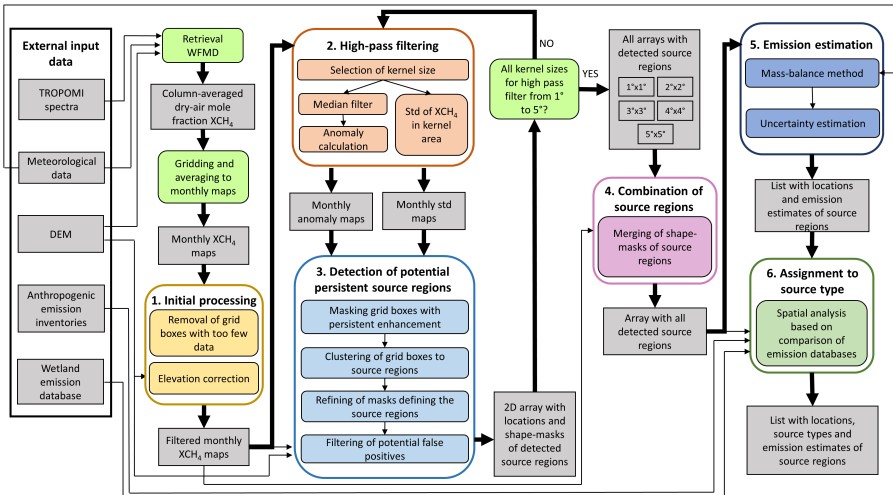

**Figure 1.** Flowchart of the Persistent Hotspot Detection (PHD) algorithm version 1.0. The colored boxes symbolize the steps in which data is processed and analyzed. The gray boxes describe the input/output data of these steps. For a detailed description of the algorithm see Sections 3.1-3.6.

## 3  Methods

We have developed a data-driven Persistent Hotspot Detection (PHD) algorithm to automatically detect regions with persistent $XCH_4$ enhancements, to estimate their emissions, and to assign a source type to these regions. The individual steps of the detection algorithm are shown in Figure 1. As input to the PHD algorithm, we use the $XCH_4$ dataset (Sect. 2.1), the wind

dataset (Sect. 2.2), the surface elevation data according to GMTED 2010 (Sect. 2.3), and the two anthropogenic emission inventories EDGAR v6.0 and GFEI v2.0, as well as the wetland emission dataset WetCHARTs v1.3.1 (Sect. 2.4). First, we process the $XCH_4$ dataset (Sect. 3.1). This step includes filtering out grid cells with too few $XCH_4$ data. For the detection of localized enhancements, we filter out large-scale $XCH_4$ variations by applying a high-pass filter with five different kernel sizes to each monthly $XCH_4$ map (Sect. 3.2), resulting in five datasets, which contain the local anomalies $\Delta XCH_4$. In the

next step, we analyze the $\Delta XCH_4$ datasets to detect persistent source regions (Sect. 3.3). For this, we first identify individual grid cells with persistent enhancement and then merge them into potential source regions. Afterwards, we conservatively filter out detected source regions, which may be false positives due to challenging surface features. For each of the five $\Delta XCH_4$ datasets, we obtain one global map of the detected potential source regions. In the next step, we combine all of the detected source regions into one map (Sect. 3.4), before we estimate their emissions (Sect. 3.5). In the final step, we determine the

dominant source types of the source regions by applying a spatial analysis based on the comparison of the methane emission databases within the source regions. As result of the PHD algorithm, we obtain a list with the characteristics of the detected source regions. The list includes the locations, the estimated emissions, and the assigned dominant source types of the source regions. In the following, we describe the steps of the algorithm in more detail.





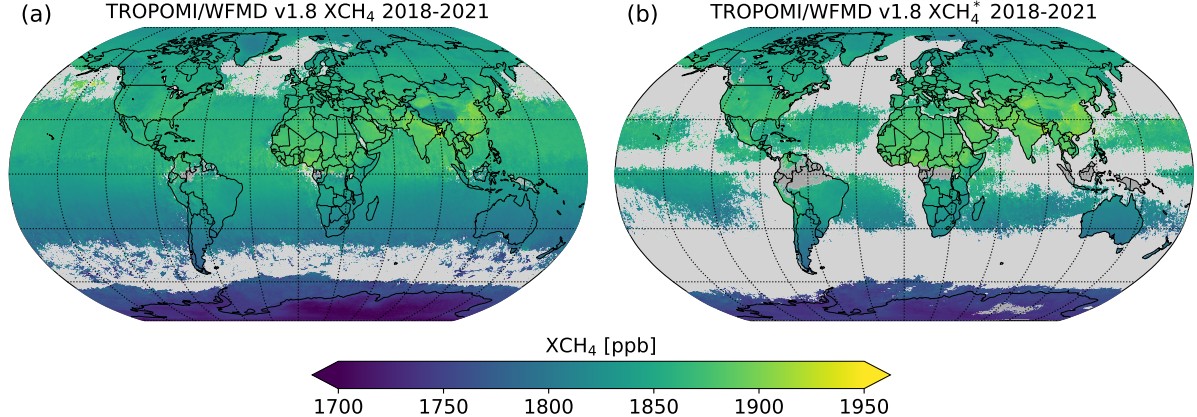

**Figure 2.** (a) Multi-year (2018-2021) XCH$_4$ and (b) the corresponding filtered and elevation corrected XCH$_4$*.

### 3.1 Initial processing

To optimize the XCH$_4$ dataset for the detection of persistent XCH$_4$ enhancements, we transform it into a new dataset XCH$_4$*. For this, we apply a filtering and a so-called elevation correction, which is described in the following. For the detection of persistent source regions, we only consider grid cells in which the monthly XCH$_4$ means were calculated from more than 3 days of TROPOMI measurements ($N_{\text{days}} > 3$).

Changes of surface elevation and tropopause height lead to variations in the tropospheric fraction of the XCH$_4$ (Kort et al.,
2014; Buchwitz et al., 2017). Because the mean mixing ratio of methane is higher in the troposphere than in the stratosphere, the XCH$_4$ over a valley is enhanced compared to its surrounding area, even if the valley is not a source region. To correct for these topography related variations, we apply an elevation correction to the XCH$_4$ (see (Buchwitz et al., 2017)). We normalize the XCH$_4$ to mean sea level by adding $8.5\,\text{ppb}$ per kilometer above mean sea level to the XCH$_4$ of the grid cells. We calculated this value by following the approach of Buchwitz et al. (2017). To determine the surface elevation of the grid cells, we use the
surface elevation data described in Sect. 2.3.

We denote the filtered and elevation corrected data as XCH$_4$*. Figure 2 shows the global maps for 2018-2021 of XCH$_4$ and XCH$_4$*. The data gaps in Fig. 2 (b) are due to the removal of the grid cells with too few data. The effect of the elevation correction can be seen in Fig. 2 (b) by higher XCH$_4$ over areas with high surface elevation (e.g. Himalaya) compared to the uncorrected dataset. In the following sections we always refer to XCH$_4$* when we mention XCH$_4$.

### 3.2 High-pass filtering

The spatial distribution of global methane concentration shows large-scale methane variations, such as the interhemispheric gradient (Figure 2 (a)). To better detect localized XCH$_4$ enhancements, we minimize these large-scale variations by applying a high-pass filter with five different kernel sizes to each monthly XCH$_4$ map (see Sect. 3.1). For each kernel size, we obtain

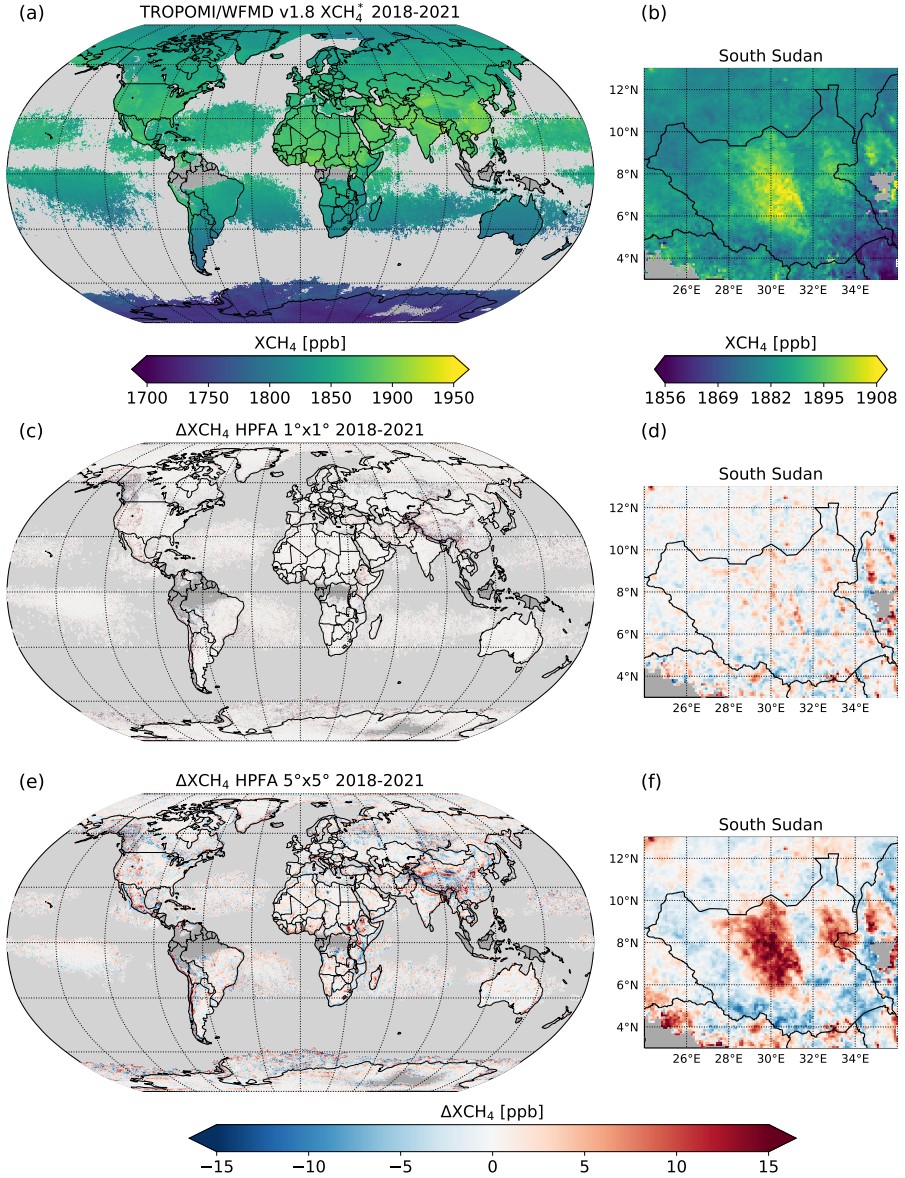

**Figure 3.** Comparison of global (left) and regional (right) multi-year (2018-2021) $XCH_4$ and $\Delta XCH_4$ maps. (a) Same as Fig. 2 (b). (b) Corresponding zoom to South Sudan. (c) $\Delta XCH_4$ calculated with a high-pass filter area (HPFA) of $1^\circ \times 1^\circ$. (d) Zoom to South Sudan of $1^\circ \times 1^\circ$ $\Delta XCH_4$ map. (e) As (c) but for a HPFA of $5^\circ \times 5^\circ$. (f) Zoom to South Sudan of $5^\circ \times 5^\circ$ $\Delta XCH_4$ map.

one dataset, which consists of monthly maps showing only the local $XCH_4$ variations. The high-pass filtering comprises three

steps and is applied to each grid cell of a monthly $XCH_4$ map as follows. First, we define an area of size $n^\circ \times n^\circ$ around the considered grid cell, denoted as high-pass filter area (HPFA($n$)), with $n \in \{1, 2, 3, 4, 5\}$. The HPFA($n$) has to be filled with at





least 25 % data. Otherwise the considered grid cell is removed. Third, we calculate the so-called methane anomaly $\Delta \mathrm{XCH_4}$ by calculating the difference of the $\mathrm{XCH_4}$ of the grid cell with the corresponding median $\widetilde{\mathrm{XCH_4}}$ in the $\mathrm{HPFA}(n)$:

$$\Delta \mathrm{XCH_4} = \mathrm{XCH_4} - \widetilde{\mathrm{XCH_4}}|_{HPFA} \tag{1}$$

In the next sections, we use the anomalies to identify potential source regions. For this, the used $\mathrm{HPFA}(n)$ has to be larger than the source regions to contain $\mathrm{XCH_4}$ which is not enhanced. Otherwise, the anomalies only describe the variations within the source regions and not their enhancements. However, the $\mathrm{HPFA}(n)$ must not be too large as it could contain $\mathrm{XCH_4}$ that is influenced by other nearby sources. Since the potential source regions to be detected have different spatial extents, ranging from small point sources to larger-scale areas, we choose five different $\mathrm{HPFA}(n)$ sizes from $n = 1°$ to $n = 5°$ to consider

source regions with various sizes.

    Figure 3 shows two multi-year $\Delta \mathrm{XCH_4}$ maps on global and regional scales, calculated with HPFA sizes of $1°$ and $5°$ (bottom), and the corresponding $\mathrm{XCH_4}$ map (top). On the left side the global maps are shown. It can be seen that the large-scale variations have been minimized in the $\Delta \mathrm{XCH_4}$ maps. The $\Delta \mathrm{XCH_4}$ maps contain less data compared to the $\mathrm{XCH_4}$ map because grid cells are filtered out whose $\mathrm{HPFA}(n)$ does not contain the minimum number of $\mathrm{XCH_4}$ data. On the right side

of Fig. 3 we show a zoom to the South Sudan region, which is a well-known source region (Pandey et al., 2021). The strong wetland emissions of the region can be seen in the resulting $\mathrm{XCH_4}$ enhancements (Fig. 3 (b)). If we compare the anomalies calculated with different HPFA of $1°$ and $5°$ (Fig. 3 (d) and (f)), we can see that the $\mathrm{HPFA}(1°)$ is too small to detect the large-scale $\mathrm{XCH_4}$ enhancements of this source region.

    In addition to the anomalies, we calculate for each grid cell the standard deviation of the $\mathrm{XCH_4}$ in the corresponding

$\mathrm{HPFA}(n)$. With that, we can determine if an anomaly is significantly enhanced compared to the variation of the surrounding $\mathrm{XCH_4}$. To reduce the impact of local $\mathrm{XCH_4}$ enhancements on the standard deviation, we use only the $\mathrm{XCH_4}$ values of the $\mathrm{HPFA}(n)$ that are smaller than the 95th percentile of the $\mathrm{XCH_4}$ distribution. In addition, we ignore the $\mathrm{XCH_4}$ value of the grid cell for which the standard deviation is calculated.

    In total, we generate five anomaly datasets consisting of monthly $\Delta \mathrm{XCH_4}$ maps and monthly standard deviation ($\sigma$) maps,

each corresponding to one of the five selected $\mathrm{HPFA}(n)$.

### 3.3   Detection of persistent potential source regions

In the third step of the PHD algorithm, we identify regions with persistent $\Delta \mathrm{XCH_4}$ enhancement in each of the five anomaly datasets calculated in Sect. 3.2. We refer to these regions as potential persistent source regions (PPSRs). To detect PPSRs, we apply the following steps:

1. We mask grid cells with persistently enhanced anomalies and cluster them to initial PPSR masks (Sect. 3.3.1)

    2. We refine the detected masks to PPSRs (Sect. 3.3.2)

    3. We filter out PPSRs with complicated surface properties (Sect. 3.3.3)

As result, for each of the five anomaly datasets, we obtain one global map containing the masks that define the PPSRs.



### 3.3.1 Detection of iPPSRs

To detect initial PPSRs (iPPSRs) in the five anomaly datasets, we proceed in two steps. First, we analyze the $\Delta$XCH$_4$ and $\sigma$ maps of each anomaly dataset to mark grid cells with persistent enhancement. Second, we cluster the detected grid cells to iPPSRs. At the end we obtain for each anomaly dataset a global map with the identified iPPSRs. To detect regions with persistent $\Delta$XCH$_4$ enhancement, we need to specify the term enhanced. Therefore, we define an anomaly as enhanced, if:

$$\Delta\text{XCH}_4 \geq N_\sigma \cdot \sigma \qquad (2)$$

We set $N_\sigma = 2$. The $\sigma$ is the standard deviation of the XCH$_4$ in the HPFA$(n)$ around the analyzed grid cell (Sect. 3.2).

To detect an iPPSR, we apply the following steps to each grid cell of the considered anomaly dataset. The steps are illustrated in Fig. 4. As first step, we define an area of $3 \times 3$ grid cells consisting of the grid cell itself and the directly adjacent grid cells. We then analyze the $\Delta$XCH$_4$ and $\sigma$ maps of the $3 \times 3$ area to check if the area shows a persistent enhancement. We are considering the adjacent grid cells in the analysis rather than only analyzing the single grid cell for the following reason. The $\Delta$XCH$_4$

enhancements within a persistent source region depend on the source itself and the meteorological conditions. Therefore, enhancements show a temporal and spatial variability. Consequently, the $\Delta$XCH$_4$ enhancements can occur at different grid cells of the persistent source region. To account for this in the detection process, we analyze the $\Delta$XCH$_4$ and $\sigma$ maps of multiple grid cells simultaneously rather than considering each grid cell independently. To check if a $3 \times 3$ area shows a persistent enhancement, we introduce several quantities to characterize the area. We count the number of months $N_{meas}$ in

which the $3 \times 3$ area contains at least one anomaly and the number of month $N_{enh}$ in which the $3 \times 3$ area contains at least one enhanced anomaly. Then, we calculate the fraction $F_{enh} = N_{enh}/N_{meas}$, which indicates how many of the months with at least one anomaly in the area show one enhanced anomaly. We also count for each grid cell of the $3 \times 3$ area the number of month $N_{enh}^{gc}$ in which the anomaly in the grid cell is enhanced. We define a $3 \times 3$ area as iPPSR, if:

$$F_{enh} \geq F_{enh,min}, \quad N_{meas} \geq N_{meas,min}, \quad N_{enh}^{gc} \text{ of central grid cell of } 3 \times 3 \text{ area} \geq 1 \qquad (3)$$

The parameter $F_{enh,min}$ and $N_{meas,min}$ define the lower limits of $F_{enh}$ and $N_{meas}$. We set $F_{enh,min} = 0.5$ and $N_{meas,min} = 16$. This means that an area is defined as iPPSR if it contains an anomaly in at least 16 of the 48 months and also contains an enhanced anomaly in at least half of the months, in which an anomaly is in the area. With the thresholds chosen, we ensure that an area has a sufficient number of months with (enhanced) anomalies during 2018-2021, but does not have to be enhanced in every month, so that it can show a temporal variability, as many methane sources typically do. We only consider $3 \times 3$ areas

as iPPSRs that have no complicated topography (median of surface roughness $< 80\,\text{m}$ and standard deviation of the surface elevation $< 150\,\text{m}$). To label a $3 \times 3$ area as iPPSR, we mark all grid cells within the area that show an enhanced anomaly in at least one month ($N_{enh}^{gc} \geq 1$). Thus, grid cells with $F_{enh} < 0.5$ can also be part of an iPPSR, if their enhancements contribute to the $3 \times 3$ area being marked as an iPPSR. As can be seen in Fig. 4 (c), the analysis of an area rather than a single grid cell enables the detection of source regions in which the individual grid cells show no persistent enhancement. This means, that the

enhanced anomalies need not to occur at the same grid cell every month, but can vary monthly within the area.



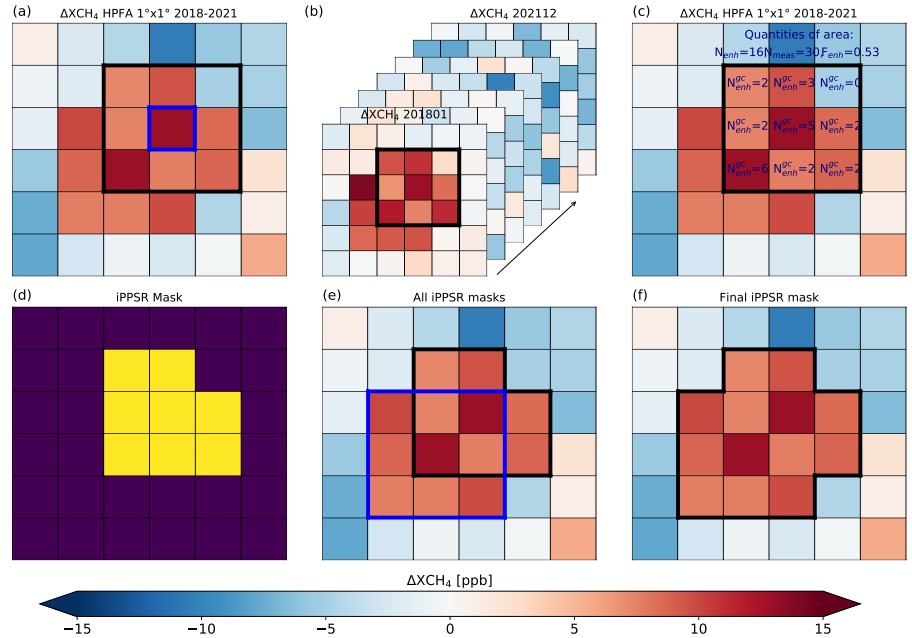

**Figure 4.** Illustration of the process to identify an initial potential persistent source region (iPPSR). (a) 2018-2021 $\Delta$XCH$_4$ calculated with a HPFA of $1° \times 1°$. The detection process comprises an analysis of the monthly $\Delta$XCH$_4$ and $\sigma$ maps for each grid cell. This analysis is illustrated for the blue outlined grid cell. First, an area of $3 \times 3$ grid cells (outlined in black) is defined around the considered grid cell. (b) Next, the anomalies within the black-outlined area are analyzed for all monthly $\Delta$XCH$_4$ and $\sigma$ maps from 2018-2021. The number of months $N_{meas}$ is counted in which at least one anomaly is in the $3 \times 3$ area. In addition, the number of months $N_{enh}$ is counted in which an enhanced anomaly is in the $3 \times 3$ area. For the definition of an enhanced anomaly see Sec. 3.3.1. Furthermore, the fraction $F_{enh} = N_{enh}/N_{meas}$ is calculated. As last part of the monthly analysis, for each grid cell within the $3 \times 3$ area, the number of months $N_{enh}^{gc}$ with enhanced anomaly is counted. (c) Multi-year $\Delta$XCH$_4$ with the results from the analysis described in (b). In each grid cell of the black outlined area $N_{enh}^{gc}$ is shown. The conditions for an iPPSR from Eq. 3 are fulfilled, since $F_{enh} \geq 0.5$, $N_{meas} \geq 16$ and $N_{enh}^{gc}$ of central grid box of the $3 \times 3$ area $\geq 1$. (d) Resulting mask (yellow grid cells) of the detected iPPSR. Only the grid cells are considered for the mask, that have an enhanced anomaly in at least one month ($N_{enh}^{gc} > 0$). (e) Multi-year $\Delta$XCH$_4$ with all detected iPPSR masks in that region. The algorithm is applied to each grid cell, resulting in an additional iPPSR being detected (outlined in blue). (f) Multi-year $\Delta$XCH$_4$ with the final iPPSR mask, which is created by merging iPPSRs that are directly adjacent or overlapping.

iPPSR that are directly adjacent or overlapping are merged into one iPPSR. For this, we apply a label algorithm in which each individual iPPSR is assigned its own number, with directly adjacent or overlapping iPPSRs getting the same number. In the end, we get a global map containing the separated and labeled iPPSRs of the considered anomaly dataset.

We apply the detection process to the five anomaly datasets and obtain five global maps with the detected iPPSRs.





### 3.3.2 Refinement of iPPSR masks to PPSR masks

The detected iPPSR masks describe the locations and shapes of the corresponding source regions. However, some of the masks do not cover the entire spatial extent of the source regions. Therefore, in the next step, we refine the iPPSR masks. One example is shown in Fig. 5 (a). It can be seen that the two iPPSR masks do not contain all the grid cells that would be identified by eye as part of the source regions, because their fractions $F_{enh}$ do not exceed the threshold $F_{enh,min} = 0.5$ required for the detection (Eq. 3). These grid cells are nevertheless part of the source region, since they have a high fraction $F_{enh}$ and are located in the immediate surroundings of the source regions. To add them to the source regions, we could lower the $F_{enh,min}$ parameter. But this would imply a change of the persistence condition. To determine the total spatial extent of the source regions without changing the persistence condition, we choose the following approach. We add grid cells to the iPPSR masks that are in the immediate vicinity and whose fractions $F_{enh}$ indicate that they are part of the source. For this, we identify all grid cells with $F_{enh} \geq 0.33$ that also fulfill all other conditions from Eq. 3. We refer to these grid cells as toseeds. The grid cells detected with $F_{enh,min} = 0.5$ are called seeds (Fig. 5 (b)). Next, we apply a random walker algorithm (Grady, 2006) to assign the toseeds to the seeds. A random walker algorithm is an image segmentation algorithm, which can divide an image into several sections based on threshold values. A first threshold is used to define the pixels of the image that represent the foreground of the image and are called seeds (the grid cells detected with $F_{enh,min}$). The seeds can have different labels, so that the foreground can be divided into different areas. With a second threshold, which is below the first one, the pixels of the background are defined which are not to be considered further. The pixels between the first and second threshold are the so called undefined pixel that the random walker algorithm assigns to the corresponding seeds by using a diffusion equation (the grid cells with $0.33 \leq F_{enh} \leq 0.5$). Based on the gradient between an undefined pixel and the different seeds and the distance between them, the probability is calculated to which seed the respective undefined pixel is assigned. The lower the gradient, i.e. the more similar the values of the undefined pixel and a seed are, the higher the probability that this pixel will be assigned to this seed. Undefined pixel that do not have a contiguous path to at least one seed are discarded. As basis on which the grid cells detected with $F_{enh,min}$ are assigned to the iPPSRs, we use the multi-year (2018-2021) $\Delta XCH_4$ of the analyzed anomaly dataset. Fig. 5 (c) shows the mask created by assigning the toseeds to the seeds. It can be seen that the spatial extent of the source regions is now better described by the masks and that grid cells are added, which connect the separate source regions. But some of the toseeds have a low multi-year $\Delta XCH_4$ mean compared to the seeds. Here, we only want to consider toseeds as part of the source region that have comparable high multi-year $\Delta XCH_4$ and remove added toseeds with $\Delta XCH_4$ smaller than $25\%$ of the maximum $\Delta XCH_4$ of the seeds. In the end, we obtain the refined iPPSR masks, which now better describe the spatial extent of the source regions and which we refer to as PPSR (Fig. 5 (d)). We emphazise, that the example shown in Fig. 5, in which two iPPSS are first merged and then separated, does not appear often. We only used it to illustrate all steps of the refinement process for one region. Due to the refinement, the number of PPSRs can differ to the number of iPPSRs. On the one hand, multiple iPSSR can be combined into one PSSR by adding new grid cells to the masks. On the other hand, an iPSSR can be split into multiple PPSRs by removing grid cells with too low 2018-2021 $\Delta XCH_4$ mean. We apply the refinement to each of the five global maps containing the detected iPPSRs (Sect. 3.3.1).





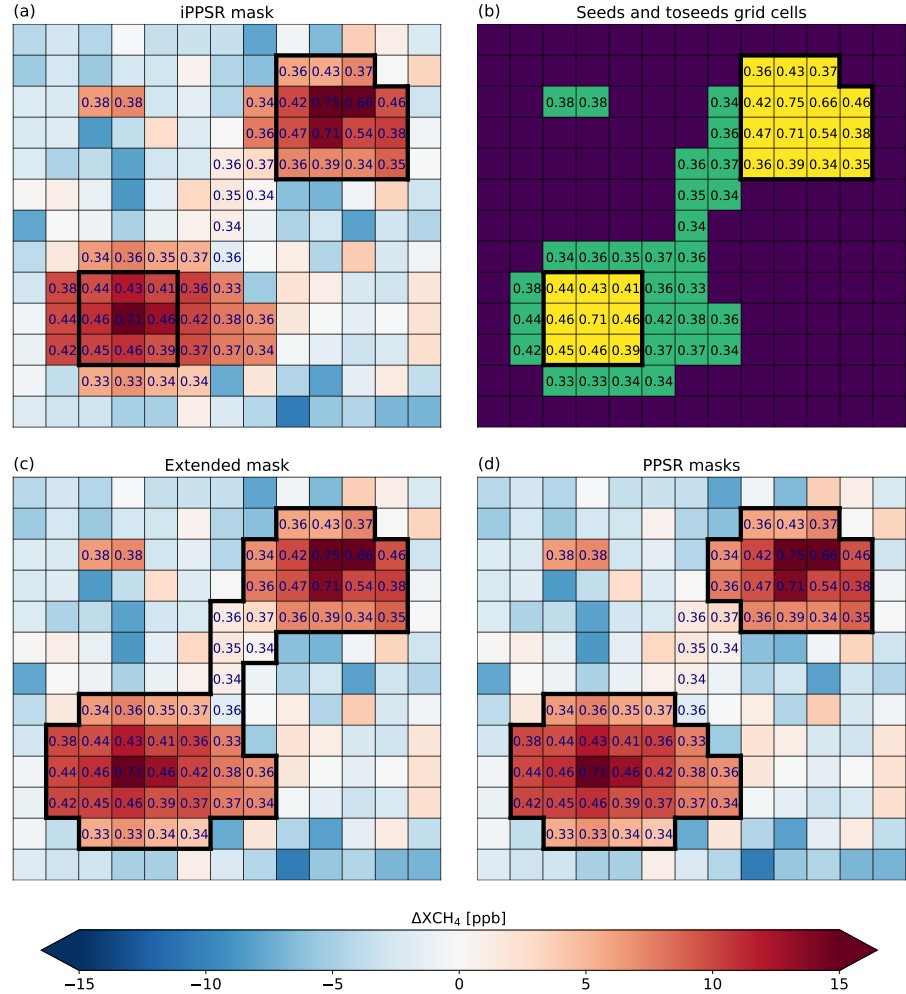

**Figure 5.** Illustration of the process to refine iPPSR masks. (a) Multi-year (2018-2021) $\Delta$XCH$_4$. In each grid cell the fraction $F_{enh}$ is shown, which is calculated for the $3 \times 3$ area of the respective grid cell (see Sect. 3.3.1). Grid cells that do not contain a fraction do not fulfill any of the conditions from Eq. 3. The detected iPPSRs (black-outlined) are the result of the detection process described in Sect. 3.3.1. Some grid cells with $F_{enh} < 0.5$ and a high multi-year $\Delta$XCH$_4$ mean would be assigned by eye as part of the source region. To add them to the mask, all grid cells are marked with $3 \times 3$ areas having a fraction $F_{enh} \geq 0.33$ and also fulfilling all other conditions from Eq. 3. These grid boxes are called toseeds and are shown in green. The grid cells of the iPPSRs (so-called seeds) are shown in yellow. (c) The toseeds are assigned to the seeds using a random walker algorithm and form together the extended mask of the iPPSRs. (d) In a final step, the grid cells with a multi-year $\Delta$XCH$_4$ mean less than $25\,\%$ of the maximum multi-year $\Delta$XCH$_4$ mean within the mask are removed from the mask. The final masks describe the refined iPPSRs and are denoted as PPSRs.



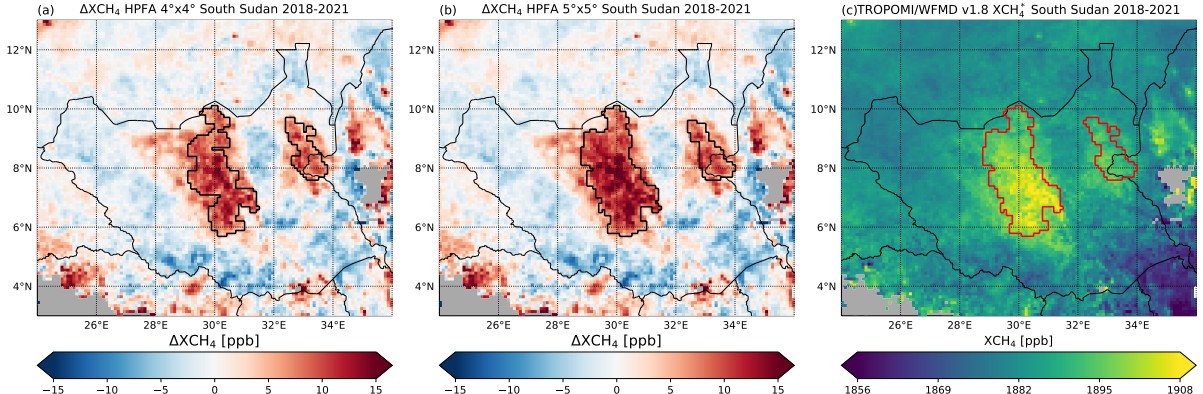

**Figure 6.** Example of PPSRs detected in two different anomaly datasets. (a) Multi-year (2018-2021) $\Delta XCH_4$ of South Sudan region calculated with a HPFA of $4° \times 4°$. The detected PPSRs (outlined in black) have already been filtered (see Sect. 3.3.3). (b) Same as (a) but for HPFA of $5° \times 5°$. (c) Corresponding 2018-2021 $XCH_4$. The final PPSR masks of the combined masks from different anomaly datasets.

### 3.3.3 Filtering of potential false positives

Much effort was made to minimize systematic biases when generating the WFMD v1.8 $XCH_4$ data product (Schneising et al., 2023). However, it is unlikely that the WFMD v1.8 product is error-free. This means that despite the good quality of the product, it is not certain that every individual $XCH_4$ enhancement has its origin in a real methane source. For example, localized $XCH_4$ enhancements could be caused by scenes with inhomogeneous albedo (e.g. coastal regions, lakes and rivers) and complex topography. To take this into account, the PPSRs are filtered for surface features, which potentially lead to a false

positive detection.. We use a conservative approach and prefer to accept false negatives rather than false positives. We decide whether a PPSR has challenging surface features based on the following properties: The correlation between SWIR surface albedo and $XCH_4$, the standard deviation of the surface elevation within the PPSR mask, the frequency of months in which the largest $XCH_4$ enhancements occur in or adjacent to grid cells with high surface roughness, the fraction of coastal grid cells in the PPSR mask and the frequency of months in which the largest $XCH_4$ enhancements occur over or next to water grid cells.

If a PPSR is identified by of these criteria then it is filtered and not considered further. Excluded from this are PPSRs in which very strong $XCH_4$ enhancements occur. By this we ensure that important source regions are not excluded due to their surface features. As we focus in this study on source regions that contribute significantly to the global methane budget, we filter out PPSRs with weak $XCH_4$ enhancements. Additionally, we filter out PPSRs that occur in the Bodélé Depression in Chad. This is a region where strong dust storms occur on average 100 days per year, always directed towards the southwest and with a

plume-like structure. Analyses of the WFMD data product have shown that these special conditions, which only occur in this region, can lead to false positive detections. We apply the filtering to each detected PPSR of each anomaly dataset to obtain five global maps comprising the refined and filtered PPSR masks, respectively.




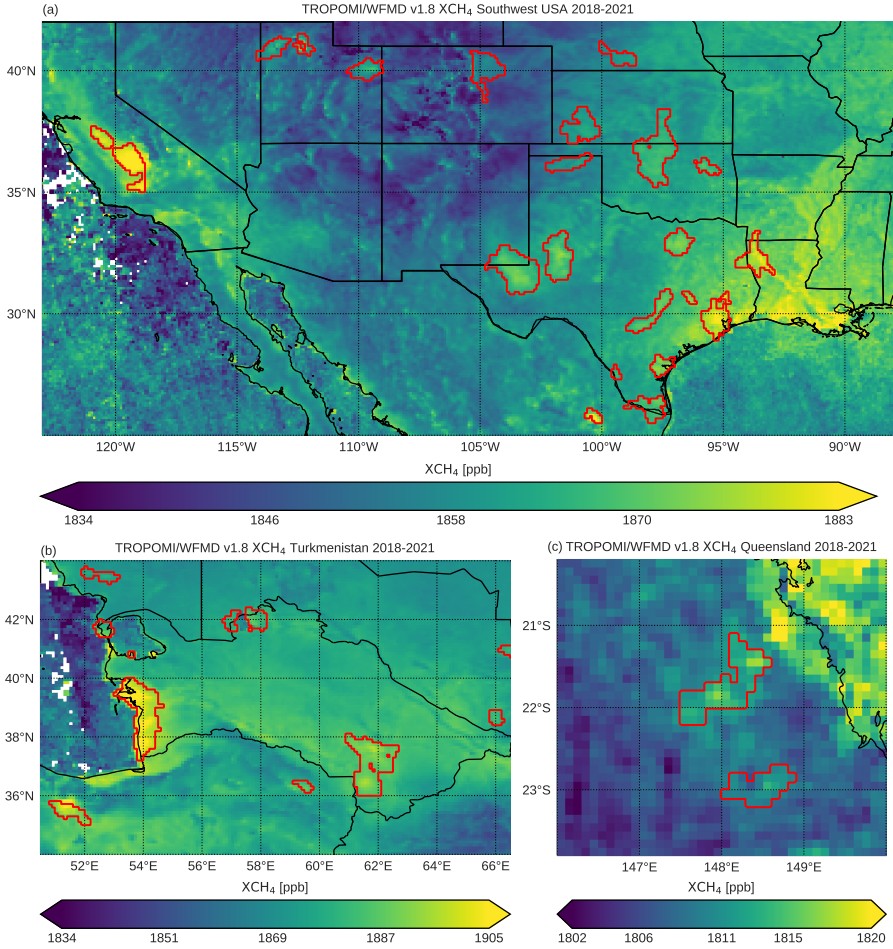

**Figure 7.** Final PPSR masks (outlined in red) after filtering (Sect. 3.3.3) and combining (Sect. 3.4) processes shown for several regions of the world. (a) 2018-2021 $XCH_4$ for the southwestern part of the USA and northern Mexico. Some of the PPSRs are located in well-known oil and gas basins like the Permian, Anadarko, Barnett, Haynesville, Denver and San Joaquin. (b) Same as (a) but for Turkmenistan, parts of Iran, Uzbekistan, and Kazakhstan. One of the detected PPSRs includes two of the largest natural gas fields in the world, Galkynish and Dauletabad. (c) Same as (a) but for parts of Queensland in Australia. Two PPSRs are detected, which are located in the Bowen Basin, a well-known coal mining area.

## 3.4 Combination of PPSRs from different anomaly datasets

We used five different $HPFA(n)$ for the calculation of the $\Delta XCH_4$ maps to detect source regions with various sizes (see Sect.

3.2). As a result, we identified different PPSRs in each anomaly dataset. To consider all PPSRs collectively, we combine them into one global map. For this, we must take into account that the same source region can be detected in multiple anomaly datasets and is thus described by more than one mask. In such a case we merge all detected masks of the PPSR to one new



mask. An example of the combination process is illustrated in Fig. 6. Here we show the well-known source regions in South Sudan (see Sect. 3.2), which we detect in the HPFA($4°$) and HPFA($5°$) anomaly datasets, and the combined masks of the

individual source regions. Finally, we obtain one global map, in which each detected source region is described by one mask. The masks of some PPSRs are shown in Fig. 7 including some well-known source regions, such as the oil and gas fields in the Permian Basin in the USA (Schneising et al., 2020; Zhang et al., 2020; Varon et al., 2023; Veefkind et al., 2023), the natural gas fields Galkynish and Dauletabad in Turkmenistan (Schneising et al., 2020) and the coal mining area in the Bowen Basin in Queensland in Australia (Sadavarte et al., 2021).

### 3.5 Emission estimation

To compute emission estimates for each of the detected PPSRs, we apply the fast data-driven method of Buchwitz et al. (2017). This method is designed to calculate averaged long-term emission estimates from time-averaged $XCH_4$ maps. It uses a conversion factor to convert an $XCH_4$ enhancement over a source region into an emission estimate. This implies the assumption that emissions from an isolated source result in an $XCH_4$ enhancement, $\delta XCH_4$, over the source region compared to the

surrounding region. To determine the monthly emission estimate $E$ (Mt yr$^{-1}$) of a PPSR, we apply the method to the monthly averaged $XCH_4$ maps using the following equation:

$$E = \delta XCH_4 \cdot M \cdot M_{exp} \cdot L \cdot V \cdot 2 \tag{4}$$

The $\delta XCH_4$ (ppb) describes the $XCH_4$ enhancement of the PPSR and is calculated by computing the difference of the mean $XCH_4$ over the source region with the mean $XCH_4$ over the surrounding region. The surrounding region is defined as described

in Fig. 8. We only consider the grid cells in the surrounding region that are not part of other PPSRs in the surrounding region. We estimate the emissions only if the PPSR, as well as the surrounding region, are each filled with at least $25\%$ data. To convert the mole fraction change $\delta XCH_4$ over the source region into a methane mass change per area, $M$ and $M_{exp}$ are used. $M$ ($5.345 \cdot 10^{-9}$ MtCH$_4$ km$^{-2}$ ppb$^{-1}$) is the methane mixing ratio enhancement to mass enhancement conversion factor for standard conditions, i.e. for a surface pressure of $1013.25$ hPa. Since the actual mass change $M_i$ of the $i$th grid cell depends

on the surface pressure $p_i$ (hPa) of the grid cell, Buchwitz et al. (2017) additionally used the dimensionless conversion factor $M_{exp}$, which is defined as:

$$M_{exp} = \frac{<M_i>}{M} \approx \frac{<p_i>}{1013.25} \approx <e^{-z_i/H}> \tag{5}$$

With surface elevation $z_i$ (km) of the $i$th grid cell, the scale height $H$ ($8.5$ km) and $<>$ denoting the mean over all grid cells of the source region. $L$ (km) in Eq. 4 is the effective length of the source region, which we calculate as the square root of the

PPSR size. $V$ (km yr$^{-1}$) is the wind speed from Sect. 2.2 averaged over the source region. The reason for adding the factor two is described in detail in Buchwitz et al. (2017), but is briefly explained in the following. When an air parcel travels with constant wind speed across the source region, it accumulates methane, which results in an $XCH_4$ enhancement when it exits the source region ($\delta XCH_{4,exit}$). However, $\delta XCH_4$ from Eq. 4 describes the mean $XCH_4$ enhancement over the source region and not $\delta XCH_{4,exit}$. Assuming a linear $XCH_4$ increase while travelling across the source region (see Fig. 3 in Buchwitz et al.





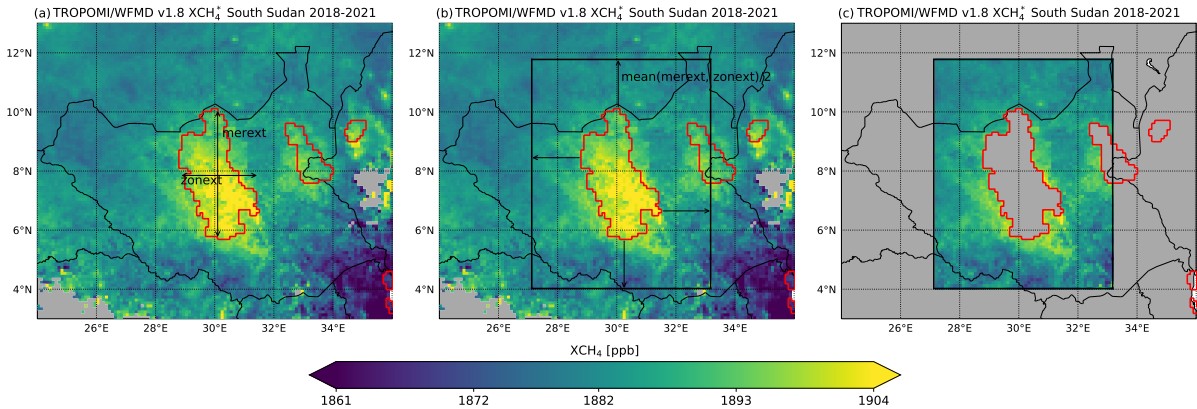

**Figure 8.** Illustration of the automated calculation of the surrounding area for a PPSR. (a) 2018-2021 $XCH_4$. The detected and unfiltered PPSRs in the HPFA($5°$) anomaly dataset for the South Sudan region are shown (outlined in red). The surrounding region for the central PPSR is calculated as follows. First the maximum extents in meridional (merext) and zonal (zonext) direction of the PPSR are calculated. (b) Next, a rectangle (black-outlined area) is defined around the PPSR by expanding the northernmost, southernmost, westernmost and easternmost coordinates by $L_{surr}$, which is half of the mean of merext and zonext. If $L_{surr}$ is smaller than $0.5°$, we set it to $0.5°$ to provide a reasonable size of the surrounding region. (c) In the last step, all grid cells outside the rectangle and all grid cells inside a source region are removed. The grid cells with $XCH_4$ are defined as the surrounding area of the central PPSR.

(2017)), these two enhancements are linked via $\delta XCH_4 = 0.5 \cdot \delta XCH_{4,exit}$. Therefore, the $\delta XCH_4$ has to be multiplied by two to describe the $XCH_4$ enhancement of the air parcel which results from the emission of the source region.

     We calculate the $1\sigma$ uncertainty of the monthly emission estimate $E$, $u_E$, by computing the sum of the squared uncertainties of the $XCH_4$ enhancement, $u_{\delta XCH_4}$, and the wind speed, $u_v$, with respect to their mean values via

$$\left(\frac{u_E}{E^2}\right) = \left(\frac{u_{\delta XCH_4}}{\delta XCH_4^2}\right) + \left(\frac{u_v}{V^2}\right) \tag{6}$$

We calculate $u_{\delta XCH_4}$ by varying the size of the surrounding region and calculating the standard deviation of the resulting $\delta XCH_4$ enhancements. We vary the region by adding to the northernmost, southernmost, westernmost, and easternmost coordinates of the surrounding region all possible combinations of 0 and $2 \times L_{surr}$, where $L_{surr}$ is the length used to define the surrounding region (see Fig. 8). The square of the uncertainty of the wind is the sum of the squared standard deviation of the monthly wind speeds within the source region, and the squared mean of the standard deviations of the wind speeds within the

months for each grid cell.

     We calculate the averaged long-term emission estimate $\overline{E}$ of a PPSR by averaging all monthly emission estimates for the period 2018-2021. For the corresponding uncertainty of the long-term emission estimate we use error propagation by computing the ratio of the root of the sum of the squared monthly uncertainties uE and the effective number of months neff





contributing to the mean estimate

$$u_{\overline{E}} = \frac{\sqrt{\sum_j u_{E,j}^2}}{n_{eff}}. \tag{7}$$

With $n_{eff}$ we consider the correlation between the monthly emission estimates. $n_{eff}$ is equal to 1 means that all emission estimates are correlated and $n_{eff}$ is equal to the total number of emission estimates means that all emission estimates are uncorrelated. We choose $n_{eff}$ with the assumption that the blocks of quarter-yearly emission estimates are uncorrelated. $n_{eff}$ is therefore the number of quarter-yearly data blocks in which at least one emission estimate contributes to the mean.

## 3.6 Assignment to source type

To determine the dominant methane source type in the detected PPSRs, we compare sector-specific emissions from different emission databases. We distinguish between the source types coal, oil and gas, other anthropogenic sources, wetlands and unknown. We use the emission data regarding coal and oil and gas from EDGAR v6.0 2018 and GFEI v2.0 2019 (Sect. 2.4). To determine the emissions originating from other anthropogenic sources, we use anthropogenic methane emissions from all sectors excluding fossil fuel from EDGAR v6.0 2018. For wetland emissions, we use the ensemble of WetCHARTs v1.3.1 for 2019 (Sect. 2.4). We assign the source type with the highest emissions as dominant source type of the corresponding PPSR. For this we sum up the emissions in the PPSR for each source type, using an expanded PPSR mask, which includes the directly adjacent outer grid cells to account for variations in the locations of the sources in the databases. We assign the type "unknown" to a PPSR if the total emissions in the respective PPSR mask are less than $50 \, \mathrm{kt} \, \mathrm{yr}^{-1}$ for all three emission databases. We choose this value by considering the PPSR with the lowest mean emission estimate for 2018-2021 ($120 \, \mathrm{kt} \, \mathrm{yr}^{-1}$) and taking into account possible differences to the emissions in the databases.

## 4 Results

In this section we present the results of the PHD algorithm, which we use to detect potential persistent source regions (PPSRs). We provide a global overview of the detected PPSRs by describing the distribution of the PPSRs among the different source types coal, oil and gas, other anthropogenic, and wetlands, as well as a rough total emission estimate of all the detected PPSRs (Sect. 4.1). We then analyze the 10 PPSRs with the highest emission estimates in more detail (Sect. 4.2). These include the Sudd Wetlands in South Sudan (Sect. 4.2.1), the west coast in Turkmenistan (Sect. 4.2.2), the Iberá wetlands in Argentina (Sect. 4.2.3), several regions in China (Sect. 4.2.4 and 4.2.5), the city Dhaka in Bangladesh and its surrounding area (Sect. 4.2.6), the Kuznetsk Basin in Russia (Sect. 4.2.7) and the Permian Basin in the United States (Sect. 4.2.8).

## 4.1 Global overview

We applied the PHD algorithm as described in Sect. 3 and detected a total of 217 PPSRs, whose global distribution and assigned source types are shown in Fig. 9. Based on the comparison of the emission databases, the fraction of dominant source types are $7.8\%$ coal, $7.8\%$ oil and gas, $30.4\%$ other anthropogenic sources, $7.3\%$ wetlands and $46.5\%$ unknown.



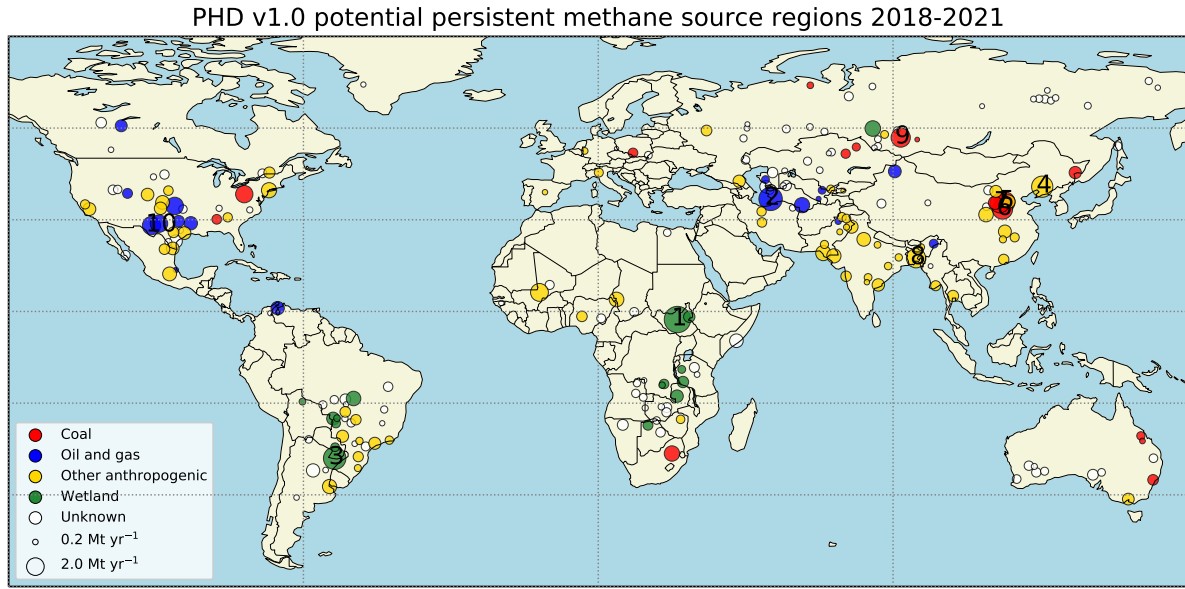

**Figure 9.** All PPSRs detected with the PHD algorithm grouped by the different dominant source types. The sizes of the circles scale with the emission estimates for 2018-2021 of the PPSRs. The 10 PPSRs with the highest emission estimates are indicated with a number.

Some of the detected source regions are well-known coal production sites, which already have been subject of several studies, such as the region Shanxi in China (Chen et al., 2022), the Bowen Basin in Queensland in Australia (Sadavarte et al., 2021), and the Upper Silesia Coal Basin in Poland (Tu et al., 2022). Other PPSRs related to coal mining activities include the Kuznetsk Basin in Russia, regions in and around Johannesburg in South Africa, the Appalachia Coal Basin in the United States, and the Ekibastuz Coal Basin in Kazakhstan. We also detect several PPSRs located in known oil and gas basins including the Permian (Schneising et al., 2020; Zhang et al., 2020; Varon et al., 2023; Veefkind et al., 2023), Uintah (de Gouw et al., 2020), Haynesville (Shen et al., 2022), and Anadarko (Schneising et al., 2020) in the USA, as well as two of the world's largest natural gas fields, Galkynish and Dauletabad in Turkmenistan (Schneising et al., 2020). A large number of the detected PPSRs are assigned to the source type other anthropogenic sources. These include regions used for agriculture, such as the Po Valley in Italy, and regions including large cities, such as Dhaka in Bangladesh, Mumbai and Delhi in India, Madrid in Spain, Buenos Aires in Argentina and Rio de Janeiro in Brazil. The emissions in these cities can originate from anthropogenic sources of different types. For example, Maasakkers et al. (2022) analyzed the methane emissions of several cities, including Mumbai, Delhi and Buenos Aires, and showed that landfills contribute to a large amount to the total emissions of these cities. In addition to anthropogenic source regions, we also detected PPSRs in wetland regions. These include well-known methane source regions like the Sudd wetlands in South Sudan (Pandey et al., 2021), the Pantanal wetlands in Brazil and the wetlands formed by the Paraná river in Argentina (Parker et al., 2018). Often, source regions contain multiple sources of different types, which is not indicated in the global map of Fig. 9. For example, we identified a source region at Lake Chad where the emission





databases indicate strong anthropogenic emissions but also strong wetland emissions. Another example is a source region in the Central Valley in the USA, which is an oil and gas production site, but also known for its livestock farming (Buchwitz et al., 2017). $46.5\%$ of the identified PPSRs are not assigned to any source type. By analyzing these in more detail, we find that most of them occur in regions with wetlands, but in which WetCHARTs v1.3.1 shows emissions lower than the used threshold of
$50\,\mathrm{kt\,yr}^{-1}$, which needs to be exceeded to assign a PPSRs to the corresponding source type (see Sect. 3.6). For example, we detected four PPSRs in Zambia, which are all known wetland methane source regions (Shaw et al., 2022), but only one of them was categorized as type wetland, while the other were assigned to type unknown. We also detected some unknown PPSRs that are located in fossil fuel production regions, such as the Cesar-Ranchería Basin in Colombia or the Surat Basin in Queensland, and some unknown PPSRs in urban areas, such as in Tulsa (USA) or in Calgary (Canada). As reported in Foy et al. (2023),
the emissions from urban areas are often underestimated in EDGAR, which may be the reason that these PPSRs could not be assigned to the other anthropogenic type.

The sum of the 2018-2021 mean emission estimates of all detected PPSRs is approximately $150\,\mathrm{Mt\,yr}^{-1}$, of which $13.0\%$ are associated with emissions from source type coal, $12.5\%$ from type oil and gas, $35.4\%$ from type other anthropogenic, $11.9\%$ from type wetland and $27.2\%$ from type unknown. We compared our total emission estimates with the calculated bottom-up
methane budget for 2017 from Saunois et al. (2020). The detected PPSRs account for $20.1\%$ of the total bottom-up emissions $(747\,\mathrm{Mt\,yr}^{-1})$, for $24.1\%$ of the emissions related to anthropogenic sources $(380\,\mathrm{Mt\,yr}^{-1})$ and for $4.9\%$ of the emissions related to natural sources $(367\,\mathrm{Mt\,yr}^{-1})$. An analysis of the anthropogenic emissions shows that the PPSRs assigned to fossil fuel account for $28.4\%$ of the total fossil fuel emissions $(135\,\mathrm{Mt\,yr}^{-1})$ reported in Saunois et al. (2020), describing $44.5\%$ of coal-related emissions $(44\,\mathrm{Mt\,yr}^{-1})$ and $22.3\%$ of oil and gas-related emissions $(84\,\mathrm{Mt\,yr}^{-1})$. The other anthropogenic PPSRs
account for $21.8\%$ of the bottom-up anthropogenic emissions that are not related to fossil fuel $(245\,\mathrm{Mt\,yr}^{-1})$. Compared to Lauvaux et al. (2022) and Schuit et al. (2023), the emissions of our detected source regions account for a larger percentage of the reported anthropogenic emissions. The detected oil and gas methane ultra-emitters by Lauvaux et al. (2022) account for $8 - 12\%$ of the oil and gas emissions reported by national inventories. In Schuit et al. (2023), anthropogenic super-emitters are detected, accounting for $2.7\%$ of the total anthropogenic emissions reported by Saunois et al. (2020). In addition to the
different methodology and data product, the higher percentage of emissions detected in our study can be explained by the focus on persistent methane sources and the additional consideration of larger-scale source regions rather than only detecting point sources.

We only detected a fraction of the global total emissions, because we only considered source regions that are localized and have a persistent enhancement, which is above a threshold. In addition, the sources can only be detected if sufficient TROPOMI
measurements are available, which depends, for example, on the presence of clouds in the considered region. Thus, emissions from sources that do not meet these criteria, such as source regions that only show strong emissions in one of the four years, cannot be detected with this method. For the calculation of the total emissions, we have to consider that a few of the detected PPSRs can be false positives, even though we applied a filtering of PPSRs in Sect. 3.3.3. If some of the PPSRs are false positives, then the calculated total emissions are overestimated.



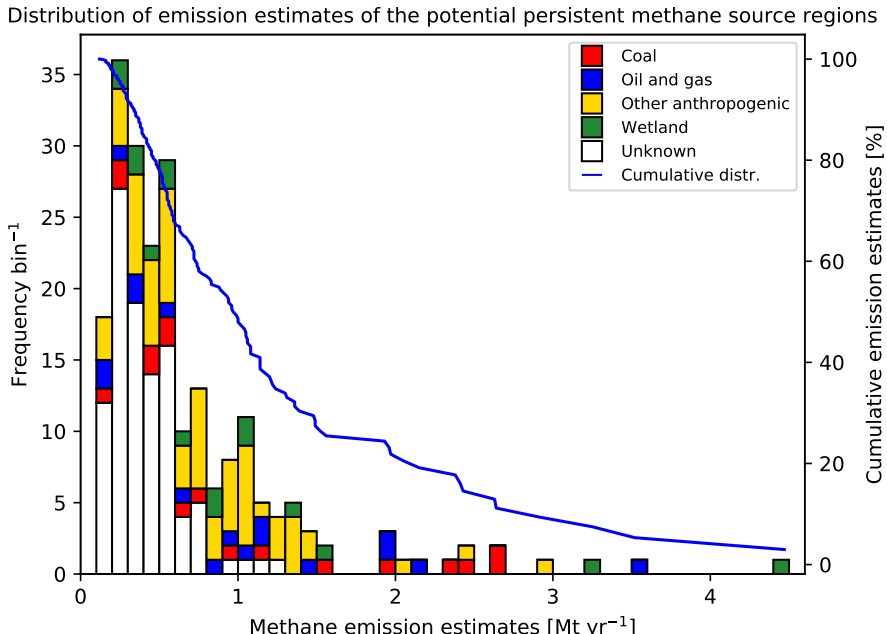

**Figure 10.** Distribution of the 2018-2021 emission estimates of the detected PPSRs, as well as the corresponding cumulative distribution (blue line). The frequency per $0.1\,\mathrm{Mt\,yr^{-1}}$ bin associated with the distribution of the emission estimates is shown on the left y-axis and the percentage share of the cumulative emission estimate of the total emission estimate is shown on the right y-axis. In each bin, the source types of the PPSRs contributing to that bin are shown with the corresponding color.

Figure 10 shows the distribution of the 2018-2021 mean emission estimates of all detected PPSRs and the corresponding cumulative distribution. The majority of the detected PPSRs, $63.6\,\%$, have a mean emission estimate between 0.1 and $0.6\,\mathrm{Mt\,yr^{-1}}$. Although the PPSRs with emission estimates greater than $0.6\,\mathrm{Mt\,yr^{-1}}$ account for only $36.4\,\%$ of the detected PPSRs, they are responsible for $66.8\,\%$ of the total detected emission estimates. Most of the PPSRs with a higher emission estimate than $0.6\,\mathrm{Mt\,yr^{-1}}$ were assigned to a source type, which indicates that the emission databases also report enhanced

methane emissions in the corresponding regions. In contrast, $64.5\,\%$ of the PPSRs with emission estimates below $0.6\,\mathrm{Mt\,yr^{-1}}$ are assigned to the unknown source type, which account for $88.1\,\%$ of all unknown PPSRs.

   For several of the detected PPSRs the emission estimates show a good agreement with emissions quantified in other studies. These include, for example, the Upper Silesia Coal Basin in Southern Poland and the Bowen Basin in Queensland in Australia. The Upper Silesia Coal Basin in Poland is one of Europe's strongest methane emission hotspots due to its intense coal mining

activities. For the PPSR in this area, we calculate an emission estimate of $0.59\pm0.11\,\mathrm{Mt\,yr^{-1}}$, which is in good agreement with emissions calculated in Tu et al. (2022) of $0.50\pm0.02\,\mathrm{Mt\,yr^{-1}}$ for the period from November 2017 to December 2020 and with emissions quantified using methane observations conducted from aircraft measurements in June 2018 during the CoMet (Carbon Dioxide and Methane Mission) campaign of $0.44\pm0.14\,\mathrm{Mt\,yr^{-1}}$ and $0.48\pm0.13\,\mathrm{Mt\,yr^{-1}}$ (Fiehn et al., 2020;



**Table 1.** Summary of the results of the 10 PPSRs with the highest methane emission estimates for 2018-2021 detected by the PHD algorithm. The $\pm$ represents the corresponding $1\sigma$ uncertainty.

| Source region | Lat. (°) | Lon. (°) | Emissions (Mt yr$^{-1}$) | XCH$_4$ (ppb) | Wind speed (m s$^{-1}$) | area (ha) | Source type |
|---|---|---|---|---|---|---|---|
| 1 South Sudan - Sudd | 7.95 | 30.15 | $4.5 \pm 0.9$ | $12.9 \pm 1.3$ | $3.9 \pm 0.6$ | 759.9 | Wetland |
| 2 Turkmenistan - Coast | 38.65 | 53.85 | $3.5 \pm 0.9$ | $17.5 \pm 1.4$ | $4.3 \pm 1.0$ | 198.3 | Oil and gas |
| 3 Argentina - Iberá | -27..35 | 302.95 | $3.3 \pm 1.0$ | $8.9 \pm 1.9$ | $5.7 \pm 1.3$ | 406.5 | Wetland |
| 4 China - Liaoning | 41.75 | 122.95 | $2.9 \pm 0.9$ | $8.2 \pm 1.6$ | $6.5 \pm 1.4$ | 290.4 | Other anthr. |
| 5 China - Shanxi 1 | 36.05 | 112.85 | $2.6 \pm 0.8$ | $25.1 \pm 2.5$ | $5.1 \pm 1.5$ | 80.0 | Coal |
| 6 China - Shanxi 2 | 37.85 | 113.45 | $2.6 \pm 0.7$ | $20.6 \pm 1.8$ | $5.9 \pm 1.3$ | 42.9 | Coal |
| 7 China - Shanxi 3 | 37.55 | 112.15 | $2.4 \pm 0.7$ | $22.3 \pm 2.5$ | $4.7 \pm 1.2$ | 63.8 | Coal |
| 8 Bangladesh - Dhaka | 23.55 | 90.85 | $2.4 \pm 0.5$ | $21.4 \pm 2.0$ | $2.9 \pm 0.6$ | 137.0 | Other anthr. |
| 9 Russia - Kuznetsk Basin | 54.25 | 86.95 | $2.4 \pm 0.5$ | $17.3 \pm 0.6$ | $4.3 \pm 0.9$ | 112.2 | Coal |
| 10 USA - Permian Delaware | 31.85 | 256.35 | $2.2 \pm 0.6$ | $7.5 \pm 0.6$ | $5.8 \pm 1.5$ | 272.9 | Oil and gas |

Fix et al., 2018). Another well-known methane source region is the Bowen Basin in Queensland in Australia, which is a coal

mining area. Here we detected two PPSRs for which the combined emission estimate is $0.63 \pm 0.16\,\mathrm{Mt\,yr^{-1}}$ for 2018-2021, which also agrees well within the uncertainties with the calculated emissions in Sadavarte et al. (2021) of $0.57 \pm 0.10\,\mathrm{Mt\,yr^{-1}}$ for 2018-2019.

## 4.2    PPSRs with highest emission estimates

An overview of the results of the 10 PPSRs with the highest emission estimates is summarized in Table 1. In the following,

each PPSR is discussed in detail, including the 2018-2021 times series for the emission estimates, XCH$_4$ enhancements and mean wind speed, and a comparison of the results with the emissions from EDGAR v6.0, GFEI v2.0, WetCHARTs v1.3.1 and related studies.

### 4.2.1    South Sudan - Sudd wetland

The PPSR with the highest emission estimate for 2018-2021, called PPSR 1, is detected in the Sudd in central South Sudan,

one of the world's largest wetlands. The South Sudan, and in particular its wetland region, is a well-known methane source region that has been subject of several studies (Frankenberg et al., 2011; Hu et al., 2018; Lunt et al., 2019; Pandey et al., 2021). By comparing the emission databases within the PPSR 1 as described in Sect. 3.6, we determine its dominant source type as wetland, which corresponds to its location in the Sudd. In Figure 11 we show an overview of the PPSR 1 results. Fig. 11 (a) shows the 2018-2021 XCH$_4$ of the South Sudan region, including the detected PPSR 1 mask, as well as one other

identified PPSR in eastern South Sudan. It can be seen that the XCH$_4$ within the PPSR 1 is strongly enhanced compared to its



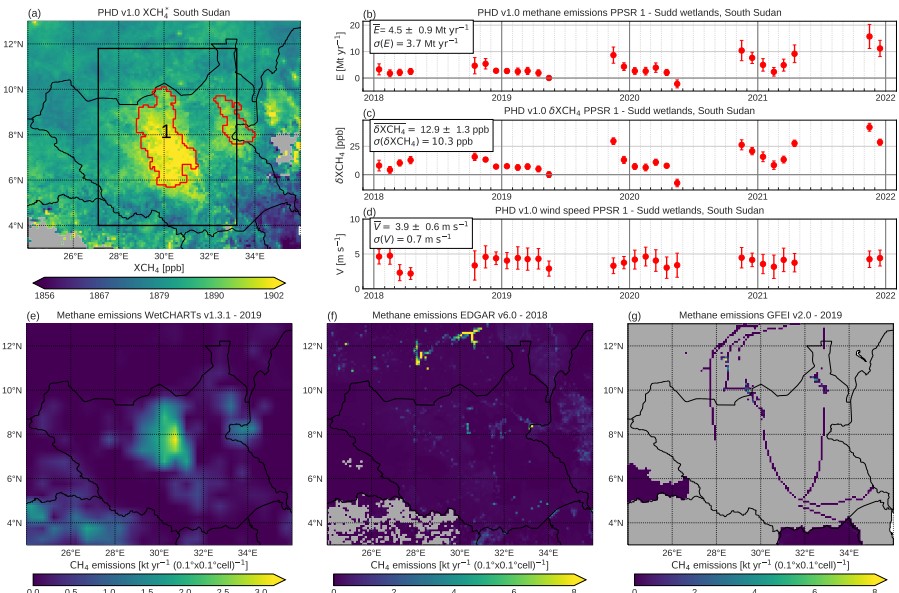

**Figure 11.** Results for the South Sudan region. (a) 2018-2021 $XCH_4$ with the detected PPSR masks outlined in red. The "1" indicates that this region is the PPSR with the highest emission estimate for 2018-2021 detected with the PHD algorithm. The black-outlined area defines the surrounding region used to calculate the $XCH_4$ enhancements $\delta XCH_4$. (b) Time series (2018-2021) of the emission estimates E, (c) $XCH_4$ enhancements $\delta XCH_4$ and (d) mean wind speed V. (e) Methane emissions from WetCHARTs v1.3.1 for 2019, (f) from EDGAR v6.0 for 2018 and (g) from GFEI v2.0 for 2019. The corresponding emissions in this PPSR are: $0.88\,\mathrm{Mt\,yr^{-1}}$ for WetCHARTs, $0.17\,\mathrm{Mt\,yr^{-1}}$ for EDGAR and $0.01\,\mathrm{Mt\,yr^{-1}}$ for GFEI.

surroundings. The area outlined in black in Fig. 11 (a) indicates the surrounding region, which is used to calculate the $XCH_4$ enhancements $\delta XCH_4$ of the PPSR 1 (see Sect. 3.5). The corresponding time series of the $\delta XCH_4$ for 2018-2021 is shown in Fig. 11 (c). The mean for the entire time period is $12.9\,\mathrm{ppb}$ with a $1\sigma$ uncertainty of $1.3\,\mathrm{ppb}$ and a standard deviation of $10.3\,\mathrm{ppb}$. The $\delta XCH_4$ shows a seasonal cycle with its peak enhancement at the end of each year, as well as an strong increase

since end of 2020. Due to frequent occurrence of clouds during the wet season from April to November, few data are available for this period of the year. In Fig. 11 (b) we show the emission estimates of PPSR 1 for 2018-2021, which we calculated as described in Sect. 3.5. The mean of the emission estimates is $4.5\pm0.9\,\mathrm{Mt\,yr^{-1}}$, where $\pm$ indicates the $1\sigma$ uncertainty. By comparing the time series in Fig. 11 (b) - (d), it can be seen, that due to the small variations of the mean wind speed $V$, the $\delta XCH_4$ variations determine the temporal variations of the emission estimates, including the strong increase since end of 2020.

This strong increase is in good agreement with the finding that tropical wetlands are a major contributor to the strong methane growth rate in 2020 and 2021 (Peng et al., 2022; Lin et al., 2023).

Pandey et al. (2021) estimated the methane emissions of the entire wetland region in South Sudan, including the Sudd and other wetlands, to be $8.0\pm3.2\,\mathrm{Mt\,yr^{-1}}$ for 2018-2019. In a study from Lunt et al. (2019) emissions of the Sudd region were estimated using GOSAT $XCH_4$ data resulting in $5.2-6.9\,\mathrm{Mt\,yr^{-1}}$ for 2016. Our estimate is lower compared to the two results,





which can be explained by the smaller source region of this study. By combining the PPSR 1 with the PPSR, which we detected in the east of South Sudan ($0.8 \pm 0.4 \, \mathrm{Mt \, yr^{-1}}$ for 2018-2021), we get the total emission estimate of $5.3 \pm 1.3 \, \mathrm{Mt \, yr^{-1}}$, which is in agreement within the uncertainties to the emissions calculated in Pandey et al. (2021) and Lunt et al. (2019).

The Emissions from the databases WetCHARTs v1.3.1, EDGAR v6.0 and GFEI v2.0 for the South Sudan region are shown in Fig. 11 (e) - (g). We compute the emissions of the databases in a PPSR by adding all emissions within the extended mask of the PPSR (see Sect. 3.6), which include the directly adjacent outer grid cells of the PPSR, to consider possible source location variations in the databases. WetCHARTs emissions for 2019 in the PPSR 1 are $0.88 \, \mathrm{Mt \, yr^{-1}}$. EDGAR's emissions for 2018 for the PPSR 1, which are mostly from the agriculture sector, combine to $0.17 \, \mathrm{Mt \, yr^{-1}}$ and the emissions from GFEI for 2019 are $0.01 \, \mathrm{Mt \, yr^{-1}}$. It can be seen, that the emissions from the databases show a large difference with the emission estimates of this study and with those of Pandey et al. (2021) and Lunt et al. (2019).

### 4.2.2 Turkmenistan - West coast

The PPSR with the second highest emission estimate for 2018-2021, called PPSR 2, is detected at the west coast of Turkmenistan, in the Balkan province, which borders the Caspian Sea. The dominant source type is determined as oil and gas. The west coast of Turkmenistan is a methane source region with oil and gas infrastructure over almost the entire coastal belt, including oil and gas power plants, compressor stations and pipelines (Irakulis-Loitxate et al., 2022). An overview of the results for PPSR 2, as well as the mask that defines the PPSR, can be seen in Fig. 12. The mean emission estimate for 2018-2021 is $3.5 \, \mathrm{Mt \, yr^{-1}}$ with an uncertainty of $0.9 \, \mathrm{Mt \, yr^{-1}}$ and a standard deviation of $0.6 \, \mathrm{Mt \, yr^{-1}}$. All months except January and February 2018 contribute to the emission estimate. The mean of the $\delta \mathrm{XCH_4}$ for the time period is $17.5 \pm 1.4 \, \mathrm{ppb}$ and the mean wind speed $4.3 \pm 1.0 \, \mathrm{m \, s^{-1}}$, where $\pm$ indicates the $1\sigma$ uncertainty.

Methane emissions on the west coast of Turkmenistan have been detected in recent studies (Irakulis-Loitxate et al., 2022; Barré et al., 2021; Schuit et al., 2023; Varon et al., 2019). In Irakulis-Loitxate et al. (2022), areas within the west coast were identified as hotspot regions using TROPOMI, where hyperspectral (ZY1 and PRISMA) and multispectral (Sentinel-2) satellites detected several localized emission events in the range of kilo tons per year from January 2017 to November 2020. In Varon et al. (2019), a methane source was detected at a compressor station in Korpezhe, in the middle of the west coast of Turkmenistan. Using TROPOMI data, the total emissions within a $12 \times 12 \, \mathrm{km^2}$ region around this source was calculated to be $0.45 \, \mathrm{Mt \, yr^{-1}} \, (0.19 - 0.75)$ for December 2017 to January 2019. The emissions calculated in these studies refer to individual events or to smaller regions of the west coast and therefore cannot be directly used for comparison with the emission estimates calculated in this study, but provide an overview of the magnitude of the emissions.

The spatial distribution of methane emissions from EDGAR v6.0 for 2018 and GFEI v2.0 for 2019 for the considered region are shown in Fig. 12 (e) - (g). The emissions from EDGAR of $0.64 \, \mathrm{Mt \, yr^{-1}}$ and GFEI of $0.62 \, \mathrm{Mt \, yr^{-1}}$ for the entire PPSR 3 are significantly lower than our estimate of $3.5 \pm 1.8 \, \mathrm{Mt \, yr^{-1}}$. Several studies suggested that the inventories may underestimate Turkmenistan's emissions (Lauvaux et al., 2022; Buchwitz et al., 2017; Shen et al., 2023). For example, Shen et al. (2023) calculated emissions of $3.6 \pm 1.3 \, \mathrm{Mt \, yr^{-1}}$ related to oil and gas in Turkmenistan using TROPOMI, which is higher as the emissions reported by GFEI of $1.5 \, \mathrm{Mt \, yr^{-1}}$. If we add the mean emission estimates of all oil and gas related PPSRs in





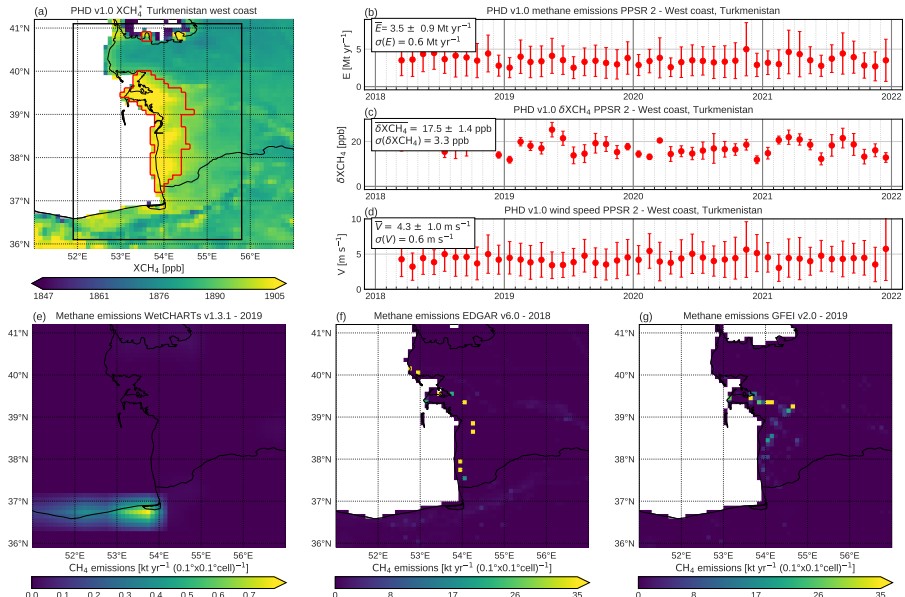

**Figure 12.** As Fig. 11 but for the west coast in Turkmenistan, where the PPSR with the second highest emission estimate for 2018-2021 is detected. The corresponding emissions of the databases in the PPSR 2 are: $0.0\,\mathrm{Mt\,yr^{-1}}$ for WetCHARTs, $0.64\,\mathrm{Mt\,yr^{-1}}$ for EDGAR and $0.62\,\mathrm{Mt\,yr^{-1}}$ for GFEI.

Turkmenistan, we get a total emission estimate of $5.0\pm1.4\,\mathrm{Mt\,yr^{-1}}$ which is in agreement within the uncertainties with Shen et al. (2023).

### 4.2.3 Argentina - Iberá wetland

The PPSR with the third highest emission estimate for 2018-2021, called PPSR 3, is detected in the region of the border between northeastern Argentina and southern Paraguay and is assigned to type wetland. The PPSR 3 is located in the northern part of the Paraná region, a well-known methane source region, which extends from the Iberá wetland in the north, the second largest wetland in the world, to the area where the Paraná river flows into the Atlantic Ocean (Parker et al., 2018). In Figure 13 we show an overview of the results of the PPSR 3. The mean emission estimate for 2018-2021 is $3.3\pm1.1\,\mathrm{Mt\,yr^{-1}}$ with a standard deviation of $1.3\,\mathrm{Mt\,yr^{-1}}$ and the mean of the corresponding $\delta\mathrm{XCH_4}$ is $8.9\pm1.9\,\mathrm{ppb}$. The emissions show a seasonal cycle, which also can be seen in the $\delta\mathrm{XCH_4}$ time series and which is in good agreement with the wet season (Ortega et al., 2022; Parker et al., 2018). Furthermore, the emission estimates show a slight decrease from 2020 onward, which agrees with the results in Lin et al. (2023), where methane emission changes between 2019 and 2021 are analyzed, including the emission changes in the Paraná region.

WetCHARTs v1.3.1 shows enhanced methane emissions for the entire Paraná region, especially for the Iberá wetland, whereas the anthropogenic databases indicate only low emissions (Fig. 13 (e) - (g)). WetCHARTs emissions for PPSR 3 are





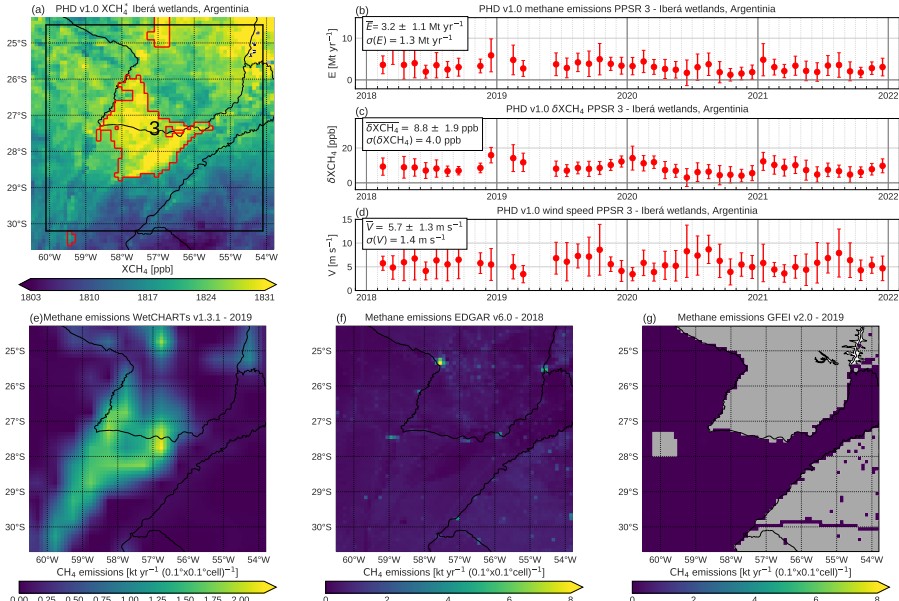

**Figure 13.** As Fig. 11 but for the Iberá wetlands in Argentina, where the PPSR with the third highest emission estimate for 2018-2021 is detected. The corresponding emissions of the databases in the PPSR 3 are: $0.64\,\mathrm{Mt\,yr^{-1}}$ for WetCHARTs, $0.18\,\mathrm{Mt\,yr^{-1}}$ for EDGAR and $0.0\,\mathrm{Mt\,yr^{-1}}$ for GFEI.

$0.64\,\mathrm{Mt\,yr^{-1}}$, which is below our emission estimate. Although the Paraná region is a known methane source region, until now,
no studies have calculated absolute values of the emissions from this region, that we can use to further assess our emission estimates. For example, in Parker et al. (2018), $XCH_4$ retrieved from GOSAT observations is used to analyze how well the methane inter-annual variability is described by model simulations for several regions, including the Paraná, without reporting explicit emission estimates.

### 4.2.4   China Liaoning

The PPSR with the fourth highest emission estimate for 2018-2021, called PPSR 4, is detected in the Liaoning province in Northeast China and is assigned to type other anthropogenic. Liaoning is known for its high agricultural production (e.g. rice cultivation and livestock) as well as for its large heavy industry, including strong coal mining activities. The results of the PPSR 4 are shown in Figure 14. The PPSR mask covers the region of the Liaoning province where most of the rice production takes place and where a majority of the coal mines are located (Ma et al., 2021; Sheng et al., 2019). The mean emission
estimate is $2.9\,\mathrm{Mt\,yr^{-1}}$ with an uncertainty of $0.9\,\mathrm{Mt\,yr^{-1}}$ and a standard deviation of $1.0\,\mathrm{Mt\,yr^{-1}}$. The $\delta XCH_4$ has a mean of $8.1\pm1.6\,\mathrm{ppb}$ and shows strong variability over the years with a standard deviation of $2.5\,\mathrm{ppb}$, with the minimum usually in spring. In all months from 2018-2021, the PPSR, as well as in the background region, are filled with sufficient $XCH_4$ values to calculate the $\delta XCH_4$ and the emission estimates.





So far, there are only a few studies that have analyzed or identified methane emissions in the considered region. For example,
two plumes were detected in 2021 by Schuit et al. (2023), which are located in the PPSR, with one plume of the dominant
source type coal and one of type landfill. In Sheng et al. (2019), coal-related emissions in 2011 for China, including the
Liaoning region, were estimated by analyzing reports from over 10000 coal mines in China. For Liaoning, the coal related
emissions were calculated to be $1.04\,\mathrm{Mt\,yr^{-1}}$. The different time periods, as well as the larger region considered in Sheng
et al. (2019) making it difficult to compare the results with the results of this study. In Foy et al. (2023) emissions of urban
areas were estimated using TROPOMI data and compared with EDGAR, including the Shenyang region in Liaoning, where
the emissions were estimated to $1.6\,\mathrm{Mt\,yr^{-1}}$. If we take into account that the Shenyang region is smaller than PPSR 5 and thus
some emissions from the surrounding area are not included in the estimate, our result is in good agreement with that of Foy
et al. (2023).

It can be seen from Fig. 14 (e) - (g), that the anthropogenic emissions are the dominant source type in this region. Emissions
from EDGAR for PPSR 4 are $1.3\,\mathrm{Mt}$ in total for 2018, with large emissions seen in Shenyang, the capital of Liaoning. Of
the $1.3\,\mathrm{Mt}$, $52\,\%$ are from the category other anthropogenic sources, which are composed of emissions from several sectors,
such as rice cultivation or landfills. The remaining emissions from EDGAR are related to the fossil fuel sector, mainly to coal
production, which is in the range of the fossil fuel related emissions from GFEI in 2019 for the PPSR of $0.49\,\mathrm{Mt\,yr^{-1}}$. The
emissions from the databases are significantly lower than the emissions calculated in this study of $2.9\pm0.9\,\mathrm{Mt\,yr^{-1}}$, which is
also reported in Foy et al. (2023) for their emission estimate of the Shenyang region.

### 4.2.5 China - Shanxi

The PPSR with the fifth, sixth and seventh highest emission estimate for 2018-2021, called PPSR 5, 6 and 7, are detected in
the Shanxi province in North China. The Shanxi province is a known methane source region with emissions resulting primarily
from high coal mining activity (Peng et al., 2023). This corresponds to the determined dominant source type of the three PPSRs,
which is coal. An overview of the results of the individual PPSRs is shown in Figure 14. Fig. 15 (a) shows the 2018-2021 $\mathrm{XCH_4}$
for Shanxi and the surroundings, including the detected PPSR masks, as well as the corresponding background regions for the
PPSR 5, 6 and 7. It can be seen that the $\mathrm{XCH_4}$ in the PPSRs is enhanced compared to the $\mathrm{XCH_4}$ in the surrounding regions.
The time series of the $\delta\mathrm{XCH_4}$ for the PPSRs are shown in Fig. 15 (c). The PPSR 5 has a mean $\delta\mathrm{XCH_4}$ for 2018-2021 of
$25.1\pm2.5\,\mathrm{ppb}$, PPSR 6 of $20.6\pm1.8\,\mathrm{ppb}$ and the PPSR 7 of $22.3\pm2.2\,\mathrm{ppb}$, which are the highest mean $\delta\mathrm{XCH_4}$ values of all
detected PPSRs. The $\delta\mathrm{XCH_4}$ shows a strong variability in all three PPSRs with standard deviations of $10.4\,\mathrm{ppb}$ for PPSR 5,
$6.5\,\mathrm{ppb}$ in PPSR 6 and $4.8\,\mathrm{ppb}$ in PPSR 7. This variability can also be seen in the emission estimates of the PPSRs shown in
Fig. 15 (b). The mean emission estimates are $2.6\pm0.8\,\mathrm{Mt\,yr^{-1}}$ for PPSR 5, $2.6\pm0.7\,\mathrm{Mt\,yr^{-1}}$ for PPSR 6 and $2.4\pm0.7\,\mathrm{Mt\,yr^{-1}}$
for PPSR 7 and in all three PPSRs almost all months contribute to the corresponding mean emission estimate.

Methane emissions in Shanxi have already been detected in several studies. The main focus was on the detection of individual
plumes, which were identified, for example, by analyzing TROPOMI data as in Schuit et al. (2023) and Lauvaux et al. (2022),
by data from the Worldview 3 satellite as in Sánchez-García et al. (2022) or by data from the PRISMA satellite mission as
in Guanter et al. (2021). The detected transient plumes in these studies are not suitable for comparison with our emission



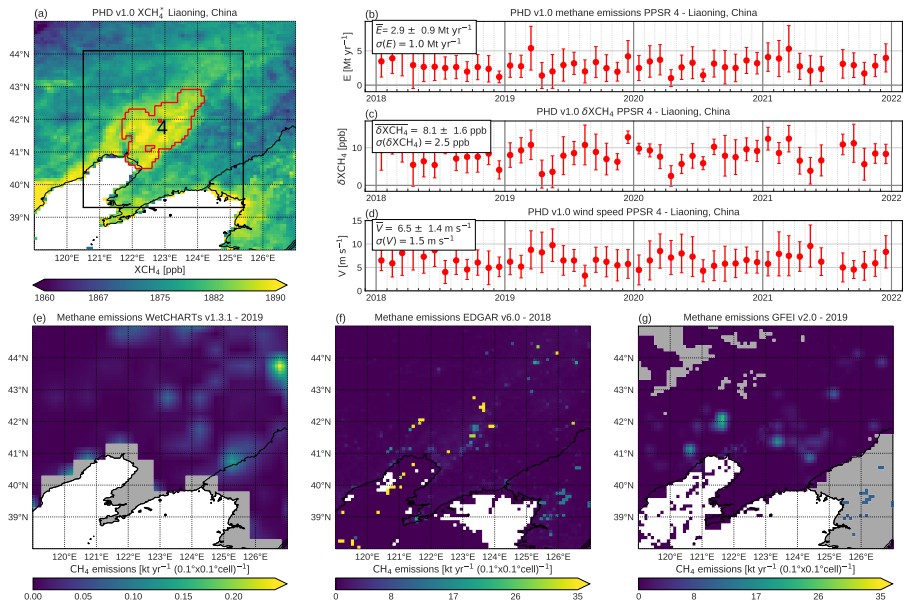

**Figure 14.** As Fig. 11 but for the Liaoning region in China, where the PPSR with the fourth highest emission estimate for 2018-2021 is detected. The corresponding emissions of the databases in the PPSR 4 are: $0.0\,\mathrm{Mt\,yr^{-1}}$ for WetCHARTs, $1.3\,\mathrm{Mt\,yr^{-1}}$ for EDGAR and $0.49\,\mathrm{Mt\,yr^{-1}}$ for GFEI.

estimates, which were evaluated for persistent hotspot regions for several years. But this is the case for the study by Peng et al. (2023), in which the coal-related methane emissions for the entire Shanxi region for the years 2019 and 2020 were

585 calculated by inversion of TROPOMI data. Peng et al. (2023) estimated emissions for 2019 of $8.5\pm0.6\,\mathrm{Mt\,yr^{-1}}$ and for 2020 of $8.6\pm0.6\,\mathrm{Mt\,yr^{-1}}$. To compare, we computed the sum of the emissions of all the detected PPSRs in Shanxi (PPSRs 5, 6, 7 and one other PPSR with a mean emission estimate of $1.1\pm0.3\,\mathrm{Mt\,yr^{-1}}$ for 2018-2021, see Fig. 15 (a)) and obtained an emission estimate of $8.8\pm2.4\,\mathrm{Mt\,yr^{-1}}$ for the period 2018-2021, which is in agreement within the uncertainties with the results from Peng et al. (2023). Moreover, by considering the emission estimates for 2019 and 2020, we obtained $8.5\pm2.1\,\mathrm{Mt\,yr^{-1}}$

for 2019 and $8.7\pm1.8\,\mathrm{Mt\,yr^{-1}}$ for 2020 for the combined PPSRs in Shanxi. In Peng et al. (2023), the entire Shanxi region is considered, while we only focused on parts of the region. However, if we assume that our identified hotspots in the Shanxi region contain the majority of methane emissions, the comparison of the two results is reasonable.

Fig. 15 (e) - (g) shows the methane emissions of WetCHARTs v1.31, EDGAR v6.0 and GFEI v2.0 for Shanxi and the surrounding area. It can be seen that the region is dominated by anthropogenic emissions. The emissions for 2018 from EDGAR

are mainly related to coal production and are $1.2\,\mathrm{Mt\,yr^{-1}}$ for PPSR 5, $2.8\,\mathrm{Mt\,yr^{-1}}$ in PPSR 6 and $1.2\,\mathrm{Mt\,yr^{-1}}$ in PPSR 7 in the corresponding extended PPSR masks. In total, the EDGAR emissions of all PPSRs in Shanxi combine to $5.2\,\mathrm{Mt\,yr^{-1}}$, which is below our emission estimate of $8.8\pm2.4\,\mathrm{Mt\,yr^{-1}}$ for 2018-2021. The emissions from GFEI for 2019 are mostly related to the coal sector and are concentrated in a few hotspots, which correlate with the locations of the detected PPSRs. For the PPSR





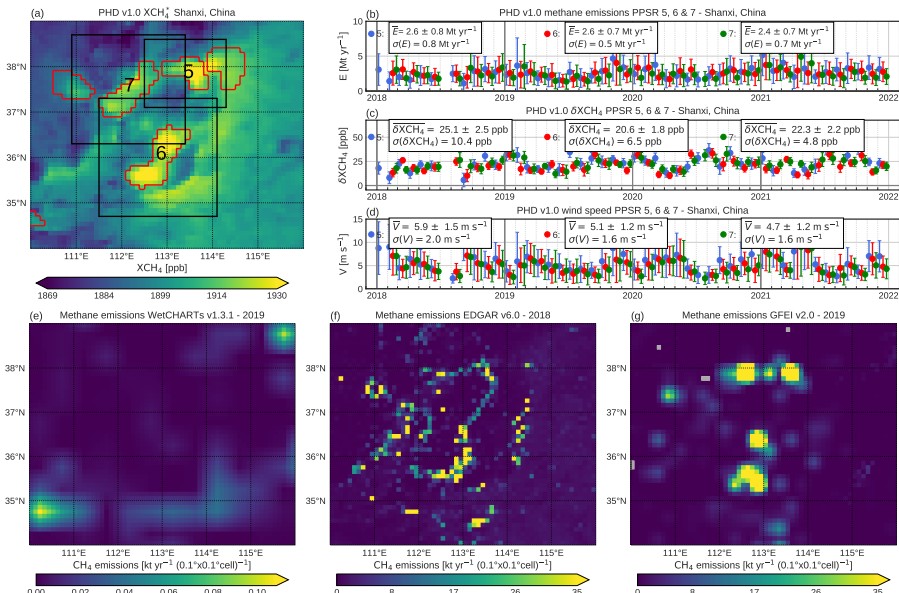

**Figure 15.** As Fig. 11 but for the Shanxi region in China, where the PPSRs with the fifth, sixth and seventh highest emission estimate for 2018-2021 is detected. The corresponding emissions of the databases in the PPSR 5, 6 and 7 are: $0\,\mathrm{Mt\,yr^{-1}}$ for WetCHARTs, $1.2\,\mathrm{Mt\,yr^{-1}}$ (PPSR 5), $2.8\,\mathrm{Mt\,yr^{-1}}$ (PPSr 6) and $1.2\,\mathrm{Mt\,yr^{-1}}$ (PPSR 7) for EDGAR and $1.5\,\mathrm{Mt\,yr^{-1}}$ (PPSR 5), $2.5\,\mathrm{Mt\,yr^{-1}}$ (PPSR 6) and $1.9\,\mathrm{Mt\,yr^{-1}}$ (PPSR 7) for GFEI.

5, the GFEI emissions are $1.5\,\mathrm{Mt\,yr^{-1}}$, $2.5\,\mathrm{Mt\,yr^{-1}}$ for PPSR 6 and $1.9\,\mathrm{Mt\,yr^{-1}}$ for PPSR 7. The total GFEI emissions of the considered PPSRs are $5.9\,\mathrm{Mt\,yr^{-1}}$, which is slightly higher as the emissions reported by EDGAR but lower than the emission estimates of this study and the study by Peng et al. (2023).

### 4.2.6 Bangladesh - Dhaka and surrounding area

The PPSR with the eighth highest emission estimate for 2018-2021, called PPSR 8, is detected in a region enclosing Dhaka, the capitol of Bangladesh, which is one of the most populated cities of the world. The dominant source type is determined as other anthropogenic sources. Dhaka and the surrounding area are a known methane source region with the main sources being agricultural production (rice, livestock) and waste management (waste water, landfills), but also with contributions from wetlands (Foy et al., 2023; Toha and Rahman, 2023). The results for the PPSR 8 are shown in Fig. 16. The 2018-2021 $XCH_4$ shows a strong enhancement in the PPSR, especially in and around Dhaka, compared to the $XCH_4$ of the surrounding area (see Fig. 16 (a)). The $\delta XCH_4$ values for 2018-2021 are shown in Fig. 16 (c), averaging to a mean of $21.4 \pm 2.0\,\mathrm{ppb}$, which is in the range of the enhancements of the PPSRs in the Shanxi region. For the considered years, no $XCH_4$ is present for the period from March/April to October/November due to the monsoon season and the resulting frequent high cloud coverage. Fig. 16 (b) shows the emission estimates for 2018-2021 with a mean of $2.4 \pm 0.5\,\mathrm{Mt\,yr^{-1}}$ and increasing values from October/November





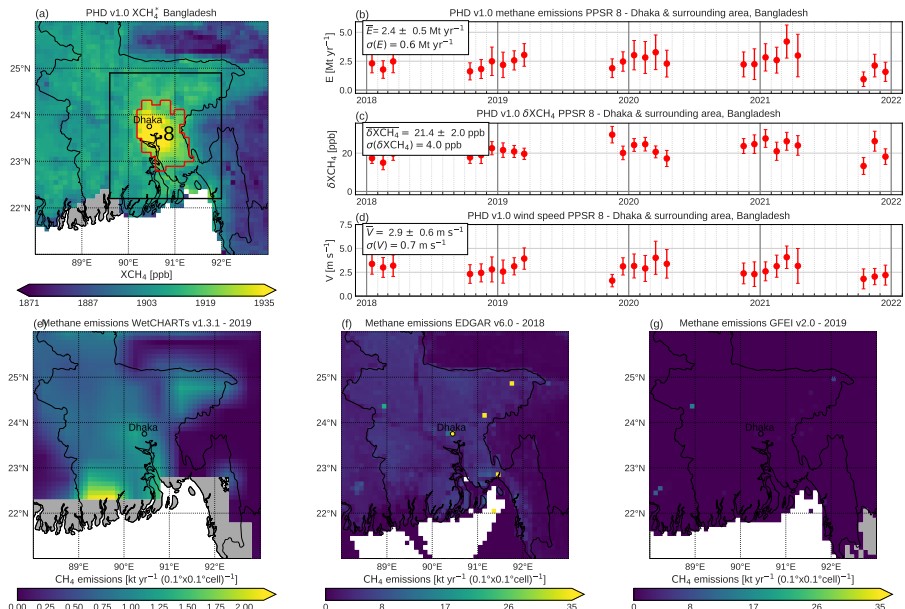

**Figure 16.** As Fig. 11 but for the region in and around Dhaka in Bangladesh, where the PPSR with the eighth highest emission estimate for 2018-2021 is detected. The corresponding emissions of the databases in the PPSR 8 are: $0.13\,\mathrm{Mt\,yr^{-1}}$ for WetCHARTs, $0.92\,\mathrm{Mt\,yr^{-1}}$ for EDGAR and $0.02\,\mathrm{Mt\,yr^{-1}}$ for GFEI.

until April/May of the following year. This period is also one of two phases, in which the rice is cultivated in Bangladesh. The first phase is in summer, which starts around June and ends in October with the harvest. The second phase is during the winter from November to April, when the fields are artificially irrigated (Rahman et al., 2023).

Methane emissions in Dhaka have already been detected and quantified in several studies (Foy et al., 2023; Schuit et al., 2023). Schuit et al. (2023) used TROPOMI data to detect plumes worldwide and detected in Dhaka as many plumes as in no other urban area. The emissions from Dhaka are calculated in Foy et al. (2023) by using TROPOMI data and a two-dimensional plume model, resulting in emissions of $1.3\,\mathrm{Mt\,yr^{-1}}$, which is lower than our estimate of $2.4 \pm 0.5\,\mathrm{Mt\,yr^{-1}}$. It must be taken into account that our region is larger than that of Foy et al. (2023) and can therefore include emissions from other cities in the surrounding area, as well as wetland emissions from the Ganges delta.

Fig. 16 (e) - (g) show the emissions from WetCHARTs v1.3.1, Edgar v6.0 and GFEI v2.0 for the Dhaka region. For WetCHARTs, the emissions in the PPSR amount to $0.13\,\mathrm{Mt}$ for 2019, for EDGAR to $0.92\,\mathrm{Mt}$ for 2018 and for GFEI to $0.02\,\mathrm{Mt}$ for 2019. The emissions from EDGAR are mainly from the agricultural sector with $0.38\,\mathrm{Mt\,yr^{-1}}$ from rice production and $0.15\,\mathrm{Mt\,yr^{-1}}$ from enteric fermentation and are lower than our calculated emission estimate. In Foy et al. (2023) the calculated emissions were also higher compared to EDGAR. They concluded that part of the difference between EDGAR and their emission estimate is due to the fact that untreated wastewater is not taken into account, which can be a major factor, especially in very densely populated cities such as Dhaka.





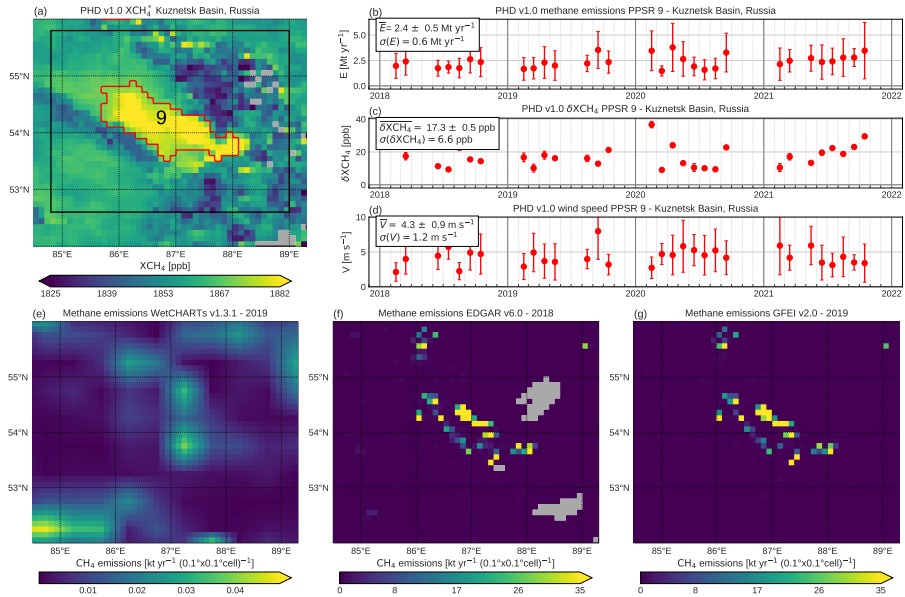

**Figure 17.** As Fig. 11 but for the Kuznetsk Basin in Russia, where the PPSR with the ninth highest emission estimate for 2018-2021 is detected. The corresponding emissions of the databases in the PPSR 9 are: $0.0\,\mathrm{Mt\,yr^{-1}}$ for WetCHARTs, $1.6\,\mathrm{Mt\,yr^{-1}}$ for EDGAR and $1.4\,\mathrm{Mt\,yr^{-1}}$ for GFEI.

### 4.2.7 Russia - Kuznetsk Basin

The PPSR with the ninth highest emission estimate for 2018-2021, called PPSR 9, is detected in the Kuznetsk Basin (also called Kuzbass) in southwestern Sibiria, Russia. Its dominant source type is determined as coal, which coincides with the fact that Kuzbass is one of the largest coal production areas worldwide (Labzovskii et al., 2022). Figure 17 shows an overview of the results for the PPSR 9. In the 2018-2021 $\mathrm{XCH_4}$ map shown in Fig. 17 (a), a strong enhancement can be seen in the entire PPSR mask compared to the $\mathrm{XCH_4}$ of the surrounding area. To quantify the $\mathrm{XCH_4}$ enhancements within the PPSR, we

computed the monthly $\delta\mathrm{XCH_4}$ for the time period 2018-2021, which are on average $17.3 \pm 0.6\,\mathrm{ppb}$ with a standard deviation of $6.6\,\mathrm{ppb}$. The mean emission estimate is $2.4\,\mathrm{Mt\,yr^{-1}}$ with an uncertainty of $0.5\,\mathrm{Mt\,yr^{-1}}$, which is computed from emission estimates of 30 months (Fig. 17 (b)).

Even though the Kuzbass is one of the largest coal production areas worldwide, there is still a need for studies reporting methane emissions from this region. In Schuit et al. (2023), methane plumes are detected in this region, but not discussed

in more detail. Due to the limited number of studies, we only compare our emission estimate with the emissions from the databases, which are shown in Figure 17 (e) - (g) for the considered region. It can be seen that the emissions from the databases are dominated by anthropogenic activity and that the emission hotspots reported by EDGAR and GFEI show a high spatial correlation. EDGAR reports emissions of $1.6\,\mathrm{Mt}$ for 2018 and GFEI of $1.4\,\mathrm{Mt}$ for 2019, whereby the emissions from both



databases are mainly related to the coal sector. Compared to the emission estimate of this study, the emissions from EDGAR
and GFEI are lower, but still within the uncertainty.

### 4.2.8   USA - Permian Basin

The PPSR with the tenth highest emission estimate for 2018-2021, called PPSR 10, is detected in the Permian Basin in the
USA and is assigned to the source type oil and gas. The Permian Basin is the most prolific oil field in the USA and also a
high producing natural gas region, which is located in western Texas and eastern New Mexico. The Permian Basin consists of
several sub-basins, including the Delaware Basin in the west and the Midland Basin in the east of the Permian, where mostly
non-conventional exploitation techniques, such as hydraulic fracturing, are used. An overview of the results for PPSR 10 are
shown in Fig. 18. It can be seen that we detect two regions in the Permian Basin. The PPSR 10 in the Delaware Basin and
a PPSR in the Midland Basin, which shows the thirteenth strongest emission estimate. Since the literature often refers to the
emissions of the entire Permian Basin, we analyze these two PPSRs together. The monthly emission estimates for 2018-2021
are shown in Fig. 18 (b). The mean emission estimate for PPSR 10 is $2.2 \pm 0.6\,\mathrm{Mt\,yr^{-1}}$ and $2.0 \pm 0.5\,\mathrm{Mt\,yr^{-1}}$ for PPSR 13,
which leads to a combined mean emission estimate of $4.1 \pm 1.1\,\mathrm{Mt\,yr^{-1}}$ for 2018-2021 (taking into account the second decimal
place). The $\delta\mathrm{XCH_4}$ time series for 2018-2021 for PPSR 10 and 13 can be seen in Fig. 18 (c). The mean $\delta\mathrm{XCH_4}$ enhancement
for PPSR 10 is $7.5 \pm 0.6\,\mathrm{ppb}$ with a standard deviation of $3.3\,\mathrm{ppb}$ and $7.2 \pm 0.6\,\mathrm{ppb}$ for PPSR 13 with a standard deviation of
$1.7\,\mathrm{ppb}$.

Methane emissions from the Permian Basins have already been quantified in several studies (Schneising et al., 2020; Shen
et al., 2022; Varon et al., 2023; Veefkind et al., 2023; Zhang et al., 2020). In the studies by Schneising et al. (2020) and Veefkind
et al. (2023), emissions were calculated based on the TROPOMI/WFMD $\mathrm{XCH_4}$ data product. Schneising et al. (2020) used
a Gaussian integral method and calculated emissions of $3.2 \pm 1.1\,\mathrm{Mt\,yr^{-1}}$ for the period 2018-2019, whereas Veefkind et al.
(2023) calculated emissions of $3.0 \pm 0.7\,\mathrm{Mt\,yr^{-1}}$ for 2019 using a divergence method. In the studies by Zhang et al. (2020),
Shen et al. (2022) and Varon et al. (2023), the emissions are calculated based on the operational TROPOMI data product and
different inversion frameworks. Zhang et al. (2020) calculated emissions of $2.9 \pm 0.5\,\mathrm{Mt\,yr^{-1}}$ for the period from May 2018 to
March 2019, whereas Shen et al. (2022) calculated emissions of $2.9 \pm 0.4\,\mathrm{Mt\,yr^{-1}}$ for the period from May 2018 to February
2020 and of $3.7 \pm 0.5\,\mathrm{Mt\,yr^{-1}}$ for the same period but with an adjusted prior. In Varon et al. (2023), the period from May
2018 to October 2020 is considered and mean emissions of $3.7 \pm 0.9\,\mathrm{Mt\,yr^{-1}}$ are calculated, which is higher than the previous
emission estimates. The emission estimate of $4.1 \pm 1.1\,\mathrm{Mt\,yr^{-1}}$ for 2018-2021 calculated in this study is slightly higher than
the emissions of the presented studies, but is in agreement within the uncertainties.

The emissions from EDGAR v6.0, GFEI v2.0 and WetCHARTs v1.3 are shown in Fig. 18 (e) - (g). For EDGAR, the
emissions within the extended PPSRs mask (see Sect. 3.6) are $1.2\,\mathrm{Mt\,yr^{-1}}$ and $0.2\,\mathrm{Mt\,yr^{-1}}$ for GFEI and relate to the oil and
gas sector. The emissions of both databases show high differences with the emission estimates of this study, as well as with the
emissions of the studies considered. In addition, the emissions of the databases also differ from one another.





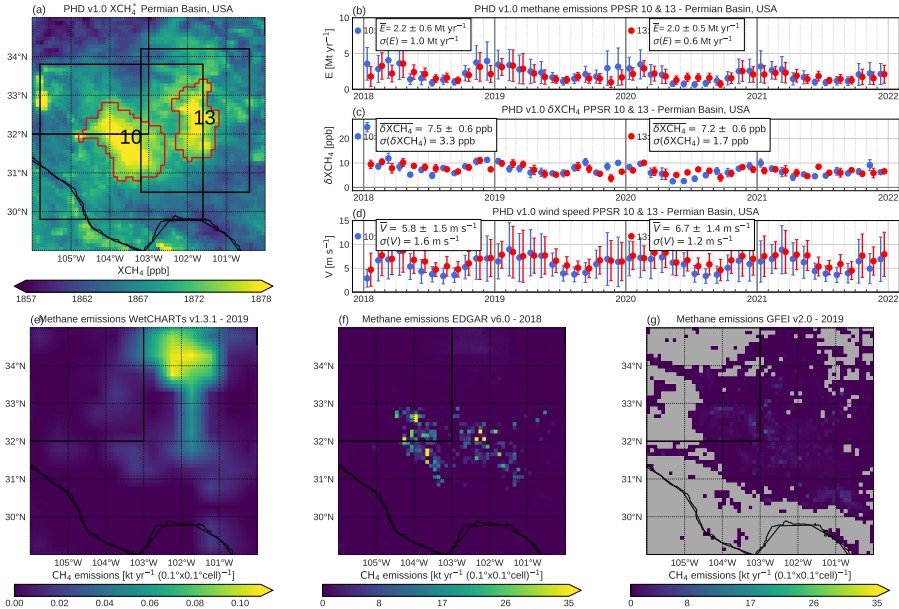

**Figure 18.** As Fig. 11 but for the Permian Basin in USA, where the PPSRs with the tenth and thirteenth highest emission estimate for 2018-2021 are detected. The corresponding emissions of the databases in the PPSR 10 and 13 are: $0.0\,\mathrm{Mt\,yr}^{-1}$ for WetCHARTs, $1.2\,\mathrm{Mt\,yr}^{-1}$ (PPSR 10) and $0.59\,\mathrm{Mt\,yr}^{-1}$ (PPSR 13) for EDGAR and $0.21\,\mathrm{Mt\,yr}^{-1}$ (PPSR 10) and $0.14\,\mathrm{Mt\,yr}^{-1}$ (PPSR 13) for GFEI.

## 5 Conclusions

We developed an automated algorithm that uses TROPOMI $XCH_4$ data to identify potential persistent methane source regions (PPSRs), to estimate their emissions and to assign a source type to them. We applied the algorithm to a dataset comprising of monthly averaged $XCH_4$ maps at $0.1° \times 0.1°$ spatial resolution from 2018-2021, which we generated by gridding the TROPOMI WFMD v1.8 data product. The detection process involves two key steps: (i) the generation of monthly methane anomaly $\Delta XCH_4$ maps, which indicate how "high" or "low" a local $XCH_4$ value is compared to the median of the surrounding $XCH_4$, and (ii) the analysis of these anomaly maps. In the letter we characterized each region by several quantities, such as the number of months in which the region shows enhanced anomalies, to then identify regions with a persistent enhancement by defining threshold values for the corresponding quantities. The algorithm is designed in a way that the thresholds can be adjusted depending on the focus of the source regions to be detected. For the automated emission estimates of the individual PPSRs, we used a fast data driven mass balance method, which is designed to calculate emission estimates from time-averaged $XCH_4$ maps. For more precise emission estimates, we recommend conducting more detailed analyses based on daily data. To determine the dominant source types of the PPSRs, we compared the emissions from several databases (WetCHARTs v1.3.1, EDGAR v6.0 and GFEI v2.0) within the PPSRs masks.

We detected a total of 217 PPSRs, of which 17 have the dominant source type coal, 17 oil and gas, 66 other anthropogenic sources, 16 wetland and 101 an unknown source type. We showed that TROPOMI data can be used to detect a variety of well-



known methane source regions such as large oil and gas fields in Turkmenistan and the USA, but also small-scale source regions like coal mines in Queensland in Australia. The emission estimates of all detected PPSRs amount to about $150\,\mathrm{Mt\,yr^{-1}}$, which corresponds to approximately $20\,\%$ of the bottom-up emissions reported in Saunois et al. (2020). We found that the majority of emissions ($35.4\,\%$) is associated with PPSRs dominated by other anthropogenic sources, followed by PPSRs of unknown type ($27.2\,\%$), type coal ($13.0\,\%$), oil and gas ($12.5\,\%$) and wetland ($11.9\,\%$). The coal-dominated source regions describe almost half ($44.5\,\%$) of global coal emissions of Saunois et al. (2020), while those from oil and gas ($22.3\,\%$), as well as other anthropogenic sources ($21.8\,\%$), also account for a large share of their sectors' emissions. This demonstrates that a comparatively small number of high-emitting source regions contributes a large proportion to the global methane emissions, underlining the importance of their detection and quantification for improving the understanding of the global methane emissions. The detected wetland regions account for $4.9\,\%$ of the total natural emissions reported in Saunois et al. (2020). However, we note that in some known wetland areas, such as Lake Chad or the Inner Niger Delta (Mali), strongly emitting PPSRs were detected, but were assigned to other source types due to the comparatively lower emissions in the wetland database. In addition, a more detailed analysis showed that many of the PPSRs with unknown source type are wetland regions. In total, $46.5\,\%$ of the PPSRs show emissions of less than $50\,\mathrm{kt\,yr^{-1}}$ in the emissions databases and were thus labeled as source regions with unknown source type. The emission estimates of the unknown PPSRs range from $0.12-1.2\,\mathrm{Mt\,yr^{-1}}$, indicating that in these regions the emission estimates of this study and the emissions in the databases have large differences. Some of the unknown PPSRs have been identified as methane sources in other studies, such as the PPSRs we detected in the Surat Basin in Australia or in the wetland region in Zambia. We found differences between the emissions of the databases and our emission estimates not only for the PPSRs with an unknown source type, but also for some of the PPSRs with the 10 highest mean emission estimates for 2018-2021. These regions are located in the Sudd wetlands in South Sudan, in the west coast of Turkmenistan, which is an area dominated by oil and gas infrastructure, the Iberá wetland in Argentina, in the Liaoning and Shanxi province in China, which are known rice and coal production areas, in the city of Dhaka and its surroundings in Bangladesh, in the Kuznetsk Basin in Russia, one of the largest coal production areas in the world, and in the Permian Basin, a large oil and gas field in the United States. For many of these PPSRs, the emission estimates are in agreement within the uncertainties with emission estimates from other studies. In the emission databases, these PPSRs are also indicated as methane hotspots, but their emissions are significantly lower compared to our emission estimates. Further studies are needed to analyze these differences between the emissions of the databases and emission estimates in this and other studies in more detail. Furthermore, we cannot exclude that some of the detected PPSRs may be false positives. To improve the filtering of potential false positives, additional parameters, such as the aerosol optical thickness, could be considered in the analysis. Since the distinction between a true and a false positive detection is not trivial in many cases, it often requires detailed analyses. For example, in Schuit et al. (2023), as well as in Lauvaux et al. (2022) human observers subsequently verify each detected plume. Such an approach was omitted in this work in order to provide a fully automated algorithm.

Each of the detected PPSRs is a potential source region that needs to be examined in more detail, for example using a similar analyses as conducted for the PPSRs with the ten highest emission estimates. Furthermore, an additional analysis of the daily data can provide new insights into the characteristics of the regions. This includes the potential to use other methods for the

calculation of the emission estimates (e.g., a gaussian integral method) or to perform detailed analyses to classify the PPSRs in terms of a false positive detection. Moreover, a more detailed comparison between the regions detected in this study and the results from the studies from Schuit et al. (2023) and Lauvaux et al. (2022), in which also methane hotspots were detected
using TROPOMI data, is of interest. The studies differ in their focus on the type of hotspot to be detected. In Schuit et al. (2023) and Lauvaux et al. (2022) the focus is on plumes originating from point sources, including short-term emissions such as gas well blowouts, while in this study persistent source regions are detected, which also include larger-scale source regions in addition to point sources. Despite these differences, a detailed comparison of these studies offers the opportunity to optimize the respective detection algorithms. The detection of known and unknown methane hotspots and the estimation of their emissions
by algorithms such as that described in this study provide important knowledge about both anthropogenic and natural sources of methane. Their operational use in the future has the potential to significantly improve the emission inventories and thus contribute to a better understanding of the evolving sources of methane in a warming world.

*Data availability.* The TROPOMI/WFMD $XCH_4$ data product is available at https://www.iup.uni-bremen.de/carbon_ghg/products/tropomi_wfmd/. EDGAR v6.0 data is available at http://data.europa.eu/89h/97a67d67-c62e-4826-b873-9d972c4f670b. GFEI v2.0 data is available
at https://doi.org/10.7910/DVN/HH4EUM. WetCHARTs v1.3.1 data is available at https://daac.ornl.gov/cgi-bin/dsviewer.pl?ds_id=1915. GMTED 2010 data are available at https://doi.org/10.5066/F7J38R2N. The ERA5 meteorological dataset is available at the Copernicus Climate Change Service (C3S) Climate Data Store (CDS) at https://cds.climate.copernicus.eu. The dataset of detected potential persistent source regions is available on request.

*Author contributions.* SV, OS and MB designed the study. SV performed the data analysis and interpreted the results with inputs from OS
and MB. SV wrote the paper with inputs from all co-authors. OS prepared the WFMD data product. HBov, HBoe, JPB and MR contributed with conceptual inputs to the improvement of the paper.

*Competing interests.* The authors have no conflict of interest to declare.

*Acknowledgements.* This publication contains modified Copernicus Sentinel data (2018-2021). Sentinel-5 Precursor is an ESA mission implemented on behalf of the European Commission. The TROPOMI payload is a joint development by the ESA and the Netherlands Space
Office (NSO). The Sentinel-5 Precursor ground-segment development has been funded by the ESA and with national contributions from the Netherlands, Germany and Belgium.



*Financial support.*  Financially support was provided in parts by the Deutsche Zentrum für Luft- und Raumfahrt (DLR) project "S5P Daten-nutzung" (grant number 50EE1811A), from the European Space Agency (ESA) via the projects GHG-CCI+ and MethaneCAMP (ESA contract nos. 4000126450/19/I-NB and 4000137895/22/I-AG), from the Federal Ministry of Education and Research (BMBF) within its project ITMS via grant no. 01 LK2103A, from the Bremer Aufbau Bank (BAB) within the project LURAFO 4009B and the University of Bremen. The TROPOMI/WFMD retrievals used in this study were performed on HPC facilities of the IUP, University of Bremen, funded under DFG/FUGG grant nos. INST 144/379-1 and INST 144/493-1.



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
