# Peer review of "Automated detection of regions with persistently enhanced methane concentrations using Sentinel-5 Precursor satellite data"

_EGUsphere, 2024_

## Author Comment (AC1)

**Finale response to referee comments on paper egusphere-2024-379**

Steffen Vanselow

July 17, 2024

**1** Answers to Reviewer 1**

We thank reviewer 1 for his/her constructive comments, which helped to improve the manuscript. Below we give answers and clarifications to all comments made by the referee (repeated after "Referee comment:"). Sentences that will appear in the revised version are shown in blue.

1. Referee comment: Abstract in L4-L8: The detailed satellite specification is not essential to include on Abstract session. Please simplify.

Authors: We simplified the corresponding sentence.

"Daily global column-averaged dry air mole fractions of atmospheric methane (XCH4) are retrieved from radiance measurements of the TROPOspheric Monitoring Instrument (TROPOMI) on board on the Sentinel-5 Precursor (S5P) satellite with a moderately high spatial resolution, enabling the detection and quantification of localized methane sources."

2. Referee comment: Introduction L34-37: This sentence is not perfect. Please re-sentence it.

Authors: We rephrased L34-L37 by splitting the two sentences in several sentences to improve readability and understanding.

"Consequently, the explanation of the observed atmospheric methane trends remains challenging. For example, the abundance of atmospheric methane grew until 1998, remained at a constant plateau until 2006, and then started to grow again. The reasons for this unique behavior are still highly debated (Nisbet et al., 2016; Turner et al., 2019). Also, the accelerated increase in recent years is still subject of ongoing research with several studies concluding that the rise was dominated by an increase in wetland emissions (Lan et al., 2021; Peng et al., 2022; Zhao et al., 2020)."

3. Referee comment: Introduction in L75-84: In this paragraph, the author identified the details of study. However, it must include and emphasize the purpose and main results through the developed automated algorithm.

Authors: To emphasize the purpose of the automated detection algorithm, we have highlighted the differences to other studies on methane source detection. We have also added a sentence including a brief overview of the methodology of the detection process. We have refrained from listing the results of the algorithm at this point, as these are already listed in the abstract and are not usually discussed in the introduction.

"The focus of the studies from Barré et al. (2021), Lauvaux et al. (2022) and Schuit et al. (2023) is on the detection of strong emitting anthropogenic points sources, for example via plume detection. But besides super-emitters, numerous larger-scale strong source regions of different source types exist, in which the emissions do not have a plumelike structure as the signals of individual sources within the regions can interfere. This can be the case, for example, in large oil and gas fields or wetlands (Lauvaux et al., 2022; Naus et al., 2023; Pandey et al., 2021). To include such source regions in a detection procedure was an important motivation for this study. Therefore, we developed an automated algorithm to detect and quantify source regions with various sizes, regardless of their source type, including small-scale super-emitters such as coal mine ventilation shafts, but also largerscale source areas such as wetland areas and large oil and gas fields. Since source regions with strong and persistent methane enhancements contribute significantly to global methane emissions, we have focussed on such source regions in this study. TROPOMI has been providing a vast amount of daily methane data since its launch in 2017. To allow the detection of methane source regions in this large dataset on a global scale, we fully automated our detection algorithm. The data-driven detection algorithm is based on several steps, including high-pass filtering of the TROPOMI data and masking of regions with persistent methane enhancements by applying different threshold criteria. In addition to detection, our algorithm includes a characterization of the source regions, in which the dominant source type is assigned and an emission estimate for each source region is determined."

4. Referee comment: Data in L102-109: For using the XCH4 by TROPOMI, why the period from 2018- to 2021 was used? In addition, for the spatial grid resolution, is 0.1 degree \* 0.1 degree statistically significant and valid? Because the TROPOMI resolution is 7\*7 km2, the 0.1-degree resolution is not sufficient for pixel sampling.

Authors: Answer to the first question: As this study focuses on the detection of persistent sources, a dataset is required that covers a period of several years. The used dataset covers 4 years from 2018-2021 and is therefore suitable for the purpose of this study. At the start of the analysis of this study, the years after 2021 had not yet been fully processed and were not included retrospectively.

Answer to the second question: The spatial grid resolution of 0.1°x0.1° is typically close to the footprint size of TROPOMI and therefore considered suitable.

In addition, other studies have shown that the resolution of  $0.1^{\circ} \times 0.1^{\circ}$  is sufficient to capture the spatial variance of XCH4 and thus to resolve the XCH4 enhancements over localized methane sources. For example, in Pandey et al. (2021) TROPOMI data is gridded to  $0.1^{\circ} \times 0.1^{\circ}$  and was used to quantify the emissions of a wetland region in South Sudan. In Shen et al. (2021) emissions of localized sources in Mexico were quantified using TROPOMI data with the spatial resolution of  $0.1^{\circ} \times 0.1^{\circ}$ . Furthermore, we showed in this study that we detect well-known source regions, which indicates that the enhancements of the source regions are well captured by the gridded XCH4 data product.

5. Referee comment: Section 2.2: How the author improve the spatial resolution for reanalysis products? It would be driven the error during the resolution improvement.

Authors: For each TROPOMI sounding the associated boundary layer wind is calculated. The resulting winds are then gridded just like  $XCH_4$  and the other relevant parameters. This will be clarified in the revised version.

As can be seen in the results in section 4.2, the monthly averaged

wind speed is the main contributor to the uncertainty of the emission estimates in most cases. The uncertainty of the monthly wind speed is calculated, in part, by calculating the standard deviation of the wind speed for the entire month. The generally high daily variability of the wind speed already leads to a large uncertainty, which is why no further consideration was given to the extent to which the error would change by adjusting the spatial resolution.

"The resulting winds are then gridded as the XCH4 dataset to monthly maps with a spatial resolution of  $0.1^{\circ} \times 0.1^{\circ}$ . In addition to the monthly averaged wind speeds, we computed the standard deviation of the wind speed within the months for each grid cell."

6. Referee comment: Section 2.4: It is too simple to explain the each datasets and its purpose for use in this study. Please improve and add the details of data characteristics and purpose of using.

Authors: For the description of the EDGAR data we added a more detailed description about the emission calculation and about the purpose of using EDGAR:

"The Emissions Database for Global Atmospheric Research (EDGAR) v6.0 (Ferrario et al., 2021) is a bottom-up inventory providing detailed information about global anthropogenic emissions of various air pollutants and greenhouse gases. The yearly emission data have a spatial resolution of  $0.1^{\circ} \times 0.1^{\circ}$  and are available from 1970 to 2018. The emissions of a specific gas are calculated using international activity data and emission factors using the IPPC (2006) methodology. Activity data describes the activities producing emissions such as the amount of fossil fuel which is exploited or the number of animals on a farm. Emission factors are coefficients that relate the emitted amount of a specific gas to a certain activity or process. The required data to calculate the emissions is collected from a variety of sources, including international organizations such as the International Energy Agency (IEA), national emission inventories and industry reports. EDGAR is well-suited to determine the anthropogenic source types of the detected potential since this inventory provides sectorspecific emissions, which enables the differentiation between individual source types within the source regions. For methane, EDGAR v6.0 provides sector-specific anthropogenic emissions from, for example, enteric fermentation, landfills, rice cultivation and fossil fuel exploitation, which are further separated into coal, oil and gas emissions. We

**use the EDGAR v6.0 methane data for 2018."**

For GFEI we also added additional information about the emission quantification and the purpose of using GFEI in addition to EDGAR.

"The Global Fuel Emission Inventory (GFEI) v2.0 (Scarpelli et al., 2022) is a methane emission database providing global anthropogenic emissions regarding the fossil fuel sectors coal, oil and gas. The emission data are gridded to yearly maps (2010-2019) with a resolution of  $0.1^{\circ} \times 0.1^{\circ}$ . GFEI v2.0 uses fossil fuel-related emission data reported by countries to the United Nations Framework Convention on Climate Change (UNFCCC), separates the emissions to sectors coal, oil and gas and assigns the data to the appropriate infrastructure locations like coal mines or oil and gas wells. The infrastructure data are taken from several databases. For countries that do not report their emissions to the UNFCCC, the emissions are calculated using IPCC (2006) methods and activity data from the US Energy Information Administration (EIA). Due to the different methods and data used for emission quantification in EDGAR v6.0 and GFEI v2.0, both databases show differences in their fossil fuel emissions, especially on a regional scale. Therefore, GFEI v2.0 can be used as a useful supplementary database to assign the appropriate fossil fuel source type to the detected source regions. We use GFEI v2.0 data for 2019."

For WetCHARTs we added the purpose of using:

"We use WetCHARTs to include also wetlands as potential dominant source type of a source region."

7. Referee comment: Figure 1: The author includes the overall of flowchart for automated algorithm in Figure 1. However, each part of the process is difficult to understand. Although the Section 3.3 explained the details of respective processes, I suggest that the Figures will add the flowchart of respective process to supplement of Section 3.3 explanation.

Authors:To make section 3.3 easier to understand, we have updated Figure 1 and added another figure (Figure 3) so that each step of the detection process now has its own figure illustrating the corresponding step.

8. Referee comment: Section 3.3: This session has to be reorganized. Some part of parameter is explained before the parameter definition, and some parts are hard to understand because of the lack of importance of used variables.

Authors: We have reorganized Section 3.3 by moving the definition of a potentially persistent source region, including the definition of all necessary parameters, to the beginning of the section. This means that all parameters are now defined before they are used.

We have also deleted the definition of an initial PPSR and referred to it directly as a PPSR in order to reduce the number of variables.

All other variables are necessary to characterize the persistence of an area. As described in Sect. 5, these parameters can be adjusted and changed as required. If the variables were omitted and replaced directly with a number (e.g. for the threshold value  $F_{enh,min}$ , which specifies the lower limit of  $F_{enh}$ ), the reader could get the impression that this is a fixed value that cannot be changed.

9. Referee comment: Caption of Figure 4 and 5: Figures' captions are too long. Because the author explained the details of figures in figures' captions, the explanation in the body of manuscript is insufficient.

Authors: We have shortened the captions of Figures 4 and 5 by removing the definition of the individual parameters (e.g.  $N_{meas}$ ,  $N_{enh}$ ) and referring to the text passages in which they are defined. Figures 4 and 5 are intended to assist in understanding the individual steps of the algorithm. For this reason, we added more references to the Figures in the explanation of the individual steps within the main text of the manuscript.

10. Referee comment: Session 3.3.2 : For the threshold selection, the author did not explain the reason. So it is hard to agree that the threshold is valid.

Authors: We added an explanation for the choice of  $F_{enh} \ge 0.33$  to the manuscript.

"We chose 0.33 as lower threshold, since  $F_{enh} \ge 0.33$  is indicating that the grid cells show enhanced anomalies in a certain number of months and are therefore still strongly influenced by the sources within the PPSR, although its  $F_{enh}$  is smaller than 0.5. Grid cells with  $F_{enh} < 0.33$  are indicating a weaker influence of the sources on the grid cells, which is why we did not include them in the refining process."

11. Referee comment: Session 3.4 in L313: During the gridding process from original pixel to gridding, is it valid to ignore the data distortion of XCH4?

Authors: No gridding takes place in the process described in section 3.4. If the referee refers to the gridding of  $XCH_4$  pixels to  $XCH_4$ maps in Sect. 2.1, this was already discussed as part of the answer to comment 4. In the answer we pointed out that other studies have shown that the spatial resolution of the gridded dataset is sufficient to capture the spatial variance of  $XCH_4$  and thus to resolve the  $XCH_4$ enhancements over localized methane sources. In addition, some of the detected PPSRs are well-known source regions, which indicates that the enhancements of these source regions are well captured with the gridded  $XCH_4$  dataset.

**12. Referee comment: Session 3.6: Please add the list of source types by Table or something.**

As suggested, we added a table with the source types. We included for each source type the used databases.

Table 1: Dominant source types of PPSRs and the corresponding databases used to estimate the sector-specific emissions.

**13. Referee comment: Table 1: For PPSR analysis, the author includes the 1sigma uncertainty. Is this a methodological error? If so, can you tell how much smaller this error is compared to other methods?**

Authors: The uncertainty in Table 1 is the uncertainty of the longterm emission estimate calculated via error propagation using the monthly 1sigma uncertainties (see Eq. 7).

A comparison of the uncertainties regarding the emission estimates calculated with other methods is not trivial. Most of the emission quantification methods such as the integrated mass enhancement (IME) or cross-sectional flux (CSF) methods calculate emissions on a daily basis, which we cannot compare to our monthly emission estimates. Therefore, we can only compare the uncertainties of the long-term emission estimates. For a detailed comparison, the emissions have to be calculated for the same region using the same data and the same time interval. For a qualitative comparison, one could compare the emission estimates of different methods, data and time intervals for the same regions, as we did in Section 4.2. Here, we have shown that for several regions, e.g. South Sudan, the long-term emissions calculated in this study show good agreement within the uncertainties with emission estimates computed in other studies.

14. Referee comment: Session 3.4 in L338: For the scale height H is assumed as 8.5 km. Do you have some references or reasons?

Authors: We refer to the scale height used in Buchwitz et al. (2017). The scale height can be calculated via H = RT/mg, with the universal gas constant R, temperature T, molar mass of dry air m and the acceleration in the gravity field of the earth g. For T = 290 K the scale height is H = 8.5 km.

**2 Answers to Reviewer 2**

We thank reviewer 2 for his/her constructive comments, which helped to improve the manuscript. Below we give answers and clarifications to all comments made by the referee (repeated after "Referee comment:"). Sentences that will appear in the revised version are shown in blue.

1. Methane concentrations are subject to long-term and short-term dynamics. For example, the research period of 2018 to 2021 contains two periods of methane trends before and after 2020. There was also a sudden change in atmospheric oxidation capacity caused by the COVID-19 pandemic. Wetland emissions show seasonality, which could also depend on latitudes. The authors identify regions with persistent enhancements based on monthly XCH4 data from all 48 months. I would expect some discussion on the impact of these dynamics on the choice of parameters (e.g., the fraction of months with enhanced anomalies) in the algorithm.

Authors: With the parameters  $F_{enh,min}$  and  $N_{meas,min}$  we determine which regions we detect as PPSRs (see Sect. 3.1.1). We have chosen

 $F_{enh,min} = 0.5$  and  $N_{meas,min} = 16$  to also consider persistent source regions in the detection process which show temporally variable emissions (e.g. wetlands) and source regions that do not have XCH4 data in every month for 2018-2021. To explain this in more detail, we added a paragraph in Sect. 3.3.1. This parameter selection enabled us to actually detect regions with temporally variable emissions (e.g., the PPSR discussed in Sect. 4.2.1).

"We have chosen  $F_{enh,min} = 0.5$  for the following reasons. Persistent methane sources do not always show enhanced methane anomalies in all months. For example, some sources show seasonal variations in emissions such as wetlands or rice paddies. Emissions from coal mines can also vary over time, as they depend on mining activity. In addition, we also want to take into account persistent sources in the detection process that started emitting during 2018-2021 and therefore do not show emissions over the entire period. With  $N_{meas,min} = 16$ , we also take into account regions that do not contain data in all 48 months."

2. The emission inventories could have large uncertainties. For example, it is kind of surprising that there are no persistent oil and gas-related source regions in Russia. It would be necessary to include some consideration about addressing the uncertainties from emission inventories in the source type assignment.

Authors: No uncertainties are specified in the databases used, which makes it difficult to address them in the source type assignment. As shown in Sect. 4.2 for some regions, EDGAR often underestimates emissions compared to emission estimates by other studies. To take this into account in the source type assignment, we have chosen the threshold value that must be exceeded by databases for the source type assignment lower than the detection limit of our method. With the threshold value, we take into account a possible underestimation by the databases, but at the same time make sure that databases show a certain amount of emissions in the assigned PPSRs. An underestimation of emissions in the databases can lead to PPSRs being marked as unknown in our algorithm. For example, both EDGAR and GFEI show emissions of a few kt yr-1 for the detected PPSRs in Russia, but these are below 50 kt yr-1, which means that these regions are assigned to the unknown type.

We added the following paragraph to Sect. 3.6.

"It should be noted that no uncertainties are specified in the used databases, which means that the uncertainties cannot be considered in the source type assignment. Therefore, we have only taken into account possible uncertainties of the databases in the sense of underestimation of emissions by setting the threshold value to be exceeded for source type assignment ( $50 \text{ kt yr}^{-1}$ ) to be significantly lower than the lowest mean emissions estimate of 2018-2021 detected by us ( $120 \text{ kt yr}^{-1}$ ). With  $50 \text{ kt yr}^{-1}$ , however, we also ensure that the databases also have a certain minimum emission when assigning a PPSR to a source type."

**3. Is there any treatment on the stripes in TROPOMI XCH4 data? How would the stripes affect the estimate of persistent enhancements?**

Authors: Version 1.8 of the XCH4 dataproduct has been optimsed in such a way that vertical stripes in the XCH4 data are removed (Schneising et al., 2023). In addition, by averaging to monthly maps, the stripes would no longer be visible in the monthly maps anyway.

4. Fig. 1: This is a very busy figure and is not very helpful for following the later explanation of the algorithm. Each step of the algorithm is complicated and requires an individual flow chart or diagram (something like Fig. 4). So I would recommend simplify this to just show the general steps.

Authors: We agree with the reviewers comment and simplified Figure 1. We also added a figure (Fig. 3 in revised manuscript) similiar to Fig. 4 (old manuscript) to explain the calculation of the methane anomalies. With that every step of the detection process of the algorithm is now illustrated with a figure to support understanding the single steps.

5. L38-42: A recent paper could fit well in this discussion about point sources: "He et al. Increased methane emissions from oil and gas following the Soviet Union's collapse. PNAS. 2024".

Authors: We thank the reviewer for the suggestion and added a reference to the paper to the manuscript.

6. L43-45: A recent paper is suitable to cite here: "Chen et al. African rice cultivation linked to rising methane. Nat. Clim. Chang. 2024".

Authors: We added a reference to the suggested paper to the manuscript.

L168 and Fig. 2: Is the threshold of Nobs>3 too low for the calculation of monthly average XCH4? It would be good to see XCH4\* filtered with other thresholds, e.g., Nobs>7, Nobs>15, etc.

Authors: We added a Fig. A1 to illustrate the differences of the monthly XCH4 maps filtered with different thresholds for  $N_{days}$ . It can be seen that with  $N_{days} > 3$ , the global coverage is higher than with larger thresholds. In combination with the threshold value  $N_{meas,min} = 16$  (see Sect. 3.3.1) which is used in the detection process and which defines the minimum number of months in which a PPSR must have measurements, it is ensured that the PPSRs have sufficient measurements to be able to draw conclusions about the XCH4 within 2018-2021. Furthermore, a higher threshold would result in less PPSRs being detected.

**8. L102-106: Is this data set related to the reprocessed XCH4 data?**

Authors: Each new data version, including version 1.8, of the TROPOMI/WFMD data product is reprocessed.

**9. L229-230: Would 3x3 grids large enough to account for meteorological effect on anomalies? This should also depend on the size of the filter, right?**

Authors: With the area of  $3 \times 3$ , we ensure that directly neighboring grid cells that show enhanced anomalies at different times are considered to be the result of the same source. The variation can be caused by meteorological variations, whereby it must be taken into account that monthly averaged XCH4 is analyzed. This means that the meteorological situations can vary strongly within a month, as a result of which the daily plumes usually average out leading to a XCH4 enhancement directly over the source region, which shows slight monthly variability. For this reason, we only consider the directly neighboring grid boxes and not a larger area. In addition, a region larger than  $3 \times 3$  is more likely to result in separate source regions being marked as one source region, especially if small localized enhancements are "smeared" in anomalies calculated with larger HP-FAs (e.g.  $5^{\circ} \times 5^{\circ}$ ). Therefore, a global criterion, independent of the HPFA suites the idea of this method.

To make this clearer in the manuscript, we have added the following paragraph to Sect. 3.3.1.

"We use an area of  $3 \times 3$  to take into account that the varying meteorological situations in the monthly XCH4 maps are not as strong as in the daily XCH4 data. In the monthly maps, the daily plumes, which vary with wind strength and direction, typically average out and result in a XCH4 enhancement over the source region, which shows only slight monthly variability."

**10. Fig. 3: Please comment on the high anomalies over the Southern Ocean and in Antarctica.**

Authors: As can be seen in Fig. A1 in the appendix, the yearly mean of the anomalies in these areas is calculated using only a few months. Therefore, these high anomalies can be the result of the high standard deviation of the few measurements. In addition, these regions will be ignored in the detection process, since  $N_{meas,min} = 16$ .

11. Section 3.2: A schematic diagram explaining the different steps in the calculation of anomalies could be helpful for an improved clarity here. It could be something similar to Figure 4.

Authors: We added a new figure (Fig. 3 in the revised manuscript) to illustrate the calculation of the anomalies using the high-pass filter.

12. Fig. 9: please add a subplot zooming in over China as there are 3 regions with top 10 emissions. Also, as the authors mentioned in the paper, would it help to add some pie charts showing the percentage of different source types as source regions could blend different types?

Authors: In Sect. 4.2 we discussed the individual regions with top 10 emissions and showed for every one of them the corresponding regions in detail, including the three regions in Shanxi, China (see Fig. 16 in the revised version). In addition, we think that zooming in over China would focus too much on this region, as clustering of strong emission regions also occurs elsewhere (e.g. USA).

Regarding the second part of the comment: We detect regions with several different source types. Among the regions with top 10 emissions, these include, for example, the PPSR in South Sudan, where the emissions from the wetland sector are  $0.88 \,\mathrm{Mt} \,\mathrm{yr}^{-1}$  and from the other anthropogenic sector  $0.16 \,\mathrm{Mt} \,\mathrm{yr}^{-1}$ , and for the PPSR in Liaoning in China, where the total emissions of EDGAR are  $1.3 \,\mathrm{Mt} \,\mathrm{yr}^{-1}$  of which 52 % are from other anthropogenic emissions and 48 % from fossil fuel (mainly coal). In many of the top 10 regions, however, there

is only one source type, as in the PPSR in Turkmenistan (oil and gas), the three PPSRs in Shanxi (coal), the PPSR in Kuznetsk Basin in Russia (coal) and the PPSR in the Permian Basin in the USA (oil and gas). If different source types occur in the regions with top 10 emissions, we have explicitly described this in the discussion of the individual regions (e.g. see Sect. 4.2.4 for Liaoning). However, as this is not the usual case, we have refrained from adding separate pie charts illustrating the emissions of the different source types in a region, but we thank the reviewer for this suggestion.

13. L440-446: The discussion should be slightly expanded here. The disproportionate distribution of methane sources agrees well with prior knowledge. Adding some references could help here, e.g., the Frankenberg et al. 2016 PNAS paper.

Authors: We added a sentence to slightly expand the discussion.

"In general, the shape of the distribution is in agreement with other studies describing a heavy-tailed distribution of strongly emitting methane emitters (Frankenberg et al., 2016; Jacob et al., 2016; Lauvaux et al., 2022; Zavala-Araiza et al., 2015)."

14. Fig. 11: The colormap makes (e), (f) and (g) hard to read.

Authors: We have changed the colors of the grid and the country borders to make the figures (e), (f) and (g) easier to read.

15. L504-512: The He et al. 2024 PNAS paper mentioned above could be added here.

Authors: We thank for the suggestion and added the named paper.

**References**

[revised manuscript text omitted]